# ACCELERATE DIFFUSION TRANSFORMERS WITH FEATURE MOMENTUM

## ABSTRACT

Diffusion models have demonstrated outstanding generative capabilities in image and video synthesis. However, their heavy computational burden, particularly due to the sequential denoising process and large model sizes, makes them challenging to meet real-time application demands. In this paper, motivated by the continuity of diffusion models in the feature space, we introduce FEMO, which employs a momentum mechanism to stabilize the dynamics of diffusion models in different timesteps, allowing us to accurately predict the features in the future timesteps based on the historical information. Additionally, we further propose an Adapted-FEMO, which allows for adaptive searching for the optimal coefficient for each generated sample. Extensive experiments demonstrate its effectiveness, *e.g.,* a **4.99× acceleration** on FLUX with **0.86 % improvements** on image reward. Under the condition of maintaining generation quality, Adapted-FEMO achieves a maximum speedup of **7.10×** on DiT and **6.24×** on FLUX. Our codes are available in the supplementary material and will be released on Github.

## 1 INTRODUCTION

In the field of generative artificial intelligence, Diffusion Models (DMs) Ho et al. (2020) have made significant progress, achieving excellent results in tasks such as image generation and video synthesis Blattmann et al. (2023); Rombach et al. (2022). Diffusion Transformers (DiT) Peebles & Xie (2023) have further improved visual generation quality by replacing the U-Net architecture with the Transformer encoder architecture. However, these advancements come with a substantial increase in computational demands, and the high-order time complexity caused by the repeated computation of high-dimensional features during inference limits the feasibility of diffusion transformers in practical applications. To address the issue of computational inefficiency, several acceleration techniques have been proposed Ma et al. (2024); Meng et al. (2023); Yuan et al. (2024); Zhao et al. (2024).

Most recently, based on the observation that diffusion models exhibit strong similarity in features between adjacent timesteps, feature caching has been proposed as a plug-and-play technique to accelerate diffusion models Selvaraju et al. (2024). Feature caching stores the features of diffusion models in previous *activated* timesteps and reuses them in the following *caching* timesteps, thus achieving significant acceleration by skipping the computation in the caching steps. Meanwhile, many studies have incorporated the characteristics of diffusion models, proposing methods such as caching important tokens Zou et al. (2024a), Zou et al. (2024b), and caching only the gap between features Chen et al. (2024). These works follow the "cache-then-reuse" paradigm that assumes that features in the previous timesteps are identical to the features in the following timesteps, which is approximately reasonable for temporally adjacent timesteps, but entirely invalid when applied to distant timesteps, leading to a significant generation quality degradation in high acceleration ratios.

**Features of diffusion models are dynamic instead of static.** More surprisingly, by visualizing the features of diffusion models in different timesteps, we find that it forms a relatively stable and continual trajectory. From observation, Liu et al. (2025) proposed TaylorSeer, a new "cache-then-forecast" paradigm that uses differential approximations of Taylor series expansions to predict the features at the current reuse step, providing a very preliminary solution to model the dynamics in the features of diffusion models. In this paper, we identify this paradigms suffer from following issues

First, Taylor-based approximations are inherently susceptible to *noise-sensitive gradient accumulation* during multi-step reuse in practical settings, as high-order derivatives such as second- or third-

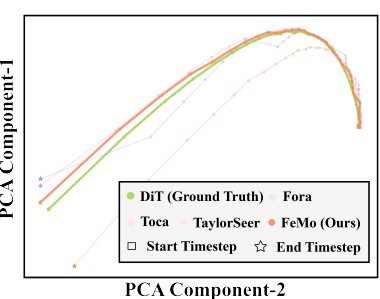 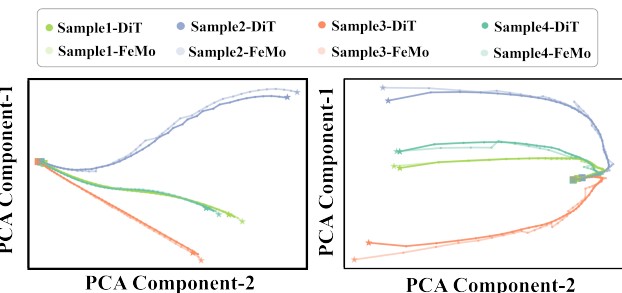

Figure 1: Scatter plot of the trajectories of FEMO and other baseline methods after PCA.

Figure 2: The trajectory of feature in diffusion models over timesteps for FEMO and the original DiT without acceleration on four different samples.

order terms are prone to estimation inaccuracies that propagate exponentially across sequential steps in the sampling process. Furthermore, the *fixed-order truncation* of Taylor series fundamentally limits its capacity to model long-term dependencies, as predefined polynomial degrees fail to account for directional reversals in feature trajectories over extended horizons. Additionally, as shown in Figure 2, there is a huge difference in the feature trajectory across timesteps for different generated samples. However, previous methods ignore their difference and treat all of them with the same paradigm. These issues introduce the requirement for a *noise-robust, long-historical, and adaptive* technique to model the dynamics of the features in diffusion models.

To address these issues, we propose Feature Momentum (FEMO) to model the dynamics in the features of diffusion models by introducing an *adaptive momentum mechanism*, allowing us to predict the features in the future timesteps based on the trend of historical features. Concretely, FEMO employs a weighted prediction mechanism, utilizing the derivative terms approximated by the differences of all fully activated timesteps to predict the features at the current reused timestep. Building on this, Adapted-FEMO minimizes the discrepancy between predicted and actual features, dynamically adjusting the weights of historical features based on the feature distribution characteristics of different samples.

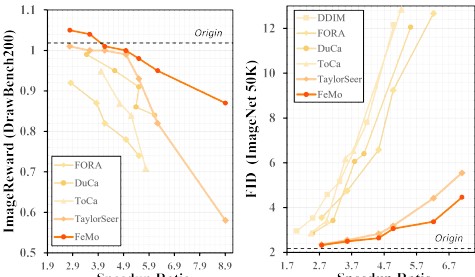

Figure 3: Comparison between the previous methods and FEMOon DrawBench with FLUX and ImageNet with DiT.

Without the requirements on high-order derivatives, FEMO avoids the high sensitivity to the outlier in diffusion progress, allowing us to precisely match the original feature trajectory of the original diffusion models at a high acceleration ratio, as shown in Figure 1. Compared to previous caching methods where high reuse frequency led to severe image quality degradation, Adapted-FEMO is particularly effective when there is a large gap between fully activated steps. As shown in Figure 8, compared with previous SOTA methods, our approach reduces quality loss by 16 times, and still maintains good generative performance under an ultra-high acceleration ratio of up to $7.1\times$, whereas previous methods experienced significant generative failure at this acceleration ratio.

In summary, the main contributions of this work are as follows:

- We propose Feature Momentum (FEMO), which predicts features of diffusion models through an adaptive momentum mechanism. The accurate prediction in FEMO allows us to skip computation in future timesteps to achieve significant acceleration without drops in generation quality.

- Based on the difference in the feature trajectory of different diffusion models, we further propose Adapted-FEMO, which dynamically adjusts the weights of historical features for each generated sample so as to effectively minimize the error caused by feature caching in practice.

- Extensive experiments on DiT and FLUX empirically demonstrate that Adapted-FEMO achieves ultra-high speedups of **6.24×** and **7.10×** respectively, while maintaining nearly lossless generation quality. It can be directly utilized in any diffusion transformer architecture without requirements for additional fine-tuning or extra training. Compared with TeaCache, FEMO achieves a **29.34%** improvement in generation quality metrics at the highest acceleration ratio.

## 2 RELATED WORK

### 2.1 DIFFUSION ACCELERATION

Since the introduction of the Diffusion model Sohl-Dickstein et al. (2015), it has made significant progress in the field of generative models, thanks largely to its exceptional capabilities in generating images and videos. The initial model primarily used the U-Net architecture Rombach et al. (2022); Ho et al. (2020), but its computational cost and inference speed bottlenecks made it difficult to meet the needs of practical applications. Although later variants like DiT Peebles & Xie (2023) enabled faster inference, they still required long overall generation times. To address this issue, various acceleration techniques have emerged in recent years, aiming to optimize both the sampling process Song et al. (2021); Lu et al. (2022b;a) and network structure Fang et al. (2024); He et al. (2024); Shang et al. (2023) of Diffusion models, in order to improve their overall generation efficiency.

**Reducing the number of sampling steps.**  DDIM Song et al. (2021) reduces the necessary sampling steps by introducing a non-Markov process, while maintaining high generation quality. In addition, some advanced higher-order ODE (ordinary differential equation) solvers, such as the DPM-Solver series Lu et al. (2022b;a), further accelerate the sampling process through more efficient numerical methods, reducing the computational load required for practical inference.

**Optimizing the computational efficiency of denoising networks.**  For example, model compression techniques such as network pruning Fang et al. (2024) and quantization He et al. (2024); Shang et al. (2023) can significantly accelerate overall inference speed without significantly reducing overall generation quality. Although these methods perform well on the U-Net architecture, there has been relatively limited systematic exploration of their application to Transformer architectures (e.g., Diffusion Transformer, DiT). Therefore, some new methods, such as FORA Selvaraju et al. (2024) and Δ-DiT Chen et al. (2024), are specifically optimized for the unique characteristics of DiT.

### 2.2 FEATURE CACHE

Feature caching technology has become an important direction for accelerating the inference of Diffusion models and has gained widespread attention in recent years. The core idea of this technology is to store and reuse intermediate features computed from previous steps during inference, in order to reduce redundant computations and improve computational efficiency. For example, DeepCache Ma et al. (2024) and Faster Diffusion Li et al. (2023) cache feature maps from intermediate layers of the U-Net model and share computational results between adjacent steps, thereby significantly reducing the computational load during inference. These methods achieve acceleration without adding extra training burdens by reducing redundant computations. However, traditional feature caching methods are primarily designed for the U-Net architecture Rombach et al. (2022); Ho et al. (2020) and are difficult to directly apply to Transformer-based Diffusion models Peebles & Xie (2023). Due to the differences in the self-attention mechanism and hierarchical structure of the Transformer architecture, traditional caching methods often fail to effectively reuse features, leading to a decline in generation quality. To address this, some new methods have proposed caching strategies specifically for Transformer architectures Ma et al. (2024); Zou et al. (2024a;b) and other related adaptive optimization algorithms Liu et al. (2024); Yuan et al. (2024); Qiu et al. (2025); Liu et al. (2025).

**Learning to optimize caching strategies.**  The Learning-to-Cache method proposed by Ma et al. Ma et al. (2024) improves caching efficiency by learning the optimal caching strategy, although this typically requires additional training steps. Additionally, ToCa Zou et al. (2024a) and DuCa Zou et al. (2024b) further reduce information loss through dynamic selection feature updates.

**Adaptive optimization strategies.**  At the same time, TeaCache Liu et al. (2024) optimizes caching decisions by dynamically selecting appropriate time steps and estimating the differences between them. DiTFastAttn Yuan et al. (2024) reduces redundancy in self-attention computations across multiple dimensions by introducing localized windowed attention, feature similarity across time steps, and the elimination of conditional redundancy. EOC Qiu et al. (2025) presents an error optimization framework that enhances caching efficiency by leveraging prior knowledge extraction

and adaptive optimization.Recently, Liu et al. (2025) proposed refining the values in the cache by approximating the true values during the next sampling step using Taylor expansion terms.

These innovative feature caching techniques provide promising new acceleration approaches for Diffusion models within the Transformer architecture. By reducing redundant computations, approximating true values in the cache, and adaptive optimization, they significantly improve inference speed while ensuring generation quality. However, these methods still face a fundamental challenge: as the time steps increase, the similarity between features rapidly decreases, leading to degradation in generation quality. Therefore, prediction-based caching methods have become a important new development trend. For example, by predicting the features of future steps, instead of directly reusing past features. Our work achieves the approximation of the "true values" during reuse in the cache with minimal additional computational cost, thus maintaining high generation quality.

## 3 METHODOLOGY

In this section, we briefly introduce the Diffusion model and Transformer Architecture, followed by Feature Caching and prediction for the Diffusion model. Then, we present the prediction principle of the **FEMO** and introduce the **Adapted-FEMO** method, which adaptively adjusts the momentum term coefficient based on the difference between predicted value and true computed value.

### 3.1 PRELIMINARY

**Diffusion model and Transformer Architecture.** The diffusion model consists of a forward process and a reverse process. The forward process gradually adds Gaussian noise to the clean image :

$$\mathbf{x}_t = \sqrt{\bar{\alpha}_t}\mathbf{x}_0 + \sqrt{1 - \bar{\alpha}_t}\epsilon \tag{1}$$

while the reverse process gradually denoises the standard Gaussian noise to recover the real image in the original data space. The denoising process is mainly based on calculating the posterior probability from the prior probability within the diffusion framework, which directly leads to the explicit probability density function of the [noised] reverse process, as defined in the following formula:

$$p_\theta(x_{t-1}|x_t) = \mathcal{N}\left(x_{t-1}; \frac{1}{\sqrt{\alpha_t}}\left(x_t - \frac{1 - \alpha_t}{\sqrt{1 - \bar{\alpha}_t}}\epsilon_\theta(x_t, t)\right), \beta_t I\right) \tag{2}$$

In this process, $T$ denotes the number of timesteps in the denoising process, $\alpha_t = 1 - \beta_t$, and $\bar{\alpha}_t = \prod_{i=1}^{T} \alpha_i$. $\epsilon_t$ represents a denoising network with inputs $\mathbf{x}_t$ and $t$. The training process involves optimizing $\theta$ such that the predicted noise removal approximates the noise added during the forward process. During image generation, the network $\epsilon_\theta$ requires $T$ inferences, consuming most of the computational cost in diffusion models. Recent studies suggest that replacing the traditional U-Net with a transformer-based architecture for $\epsilon_\theta$ can significantly enhance generation quality. Diffusion Transformer models are usually composed of stacking groups of self-attention layers $f_{SA}$, multilayer perceptron $f_{MLP}$, and cross-attention layers $f_{CA}$ (for conditional generation).The input data $x_t$ consists of a sequence of tokens representing various patches within the generated images. This can be expressed as $x_t = \{x_i\}_{i=1}^{H \times W}$, where $H$ and $W$ correspond to the height and width of the images, or the latent code of the images, respectively.

**Feature Caching and predicting for Diffusion model.** Recent acceleration techniques apply Naïve Feature Caching in diffusion models by reusing features from adjacent timesteps:

$$\mathcal{F}(x_{t-k}) := \mathcal{F}(x_t), \quad \text{where } k \in \{1, \dots, N - 1\} \tag{3}$$

This strategy theoretically provides a speedup of $(N - 1)$-times, but errorr accumulation caused by direct reuse limits maximum speedup before model failure. A new method TaylorSeer has recently been proposed. It simulates the first-order derivative using finite differences, and applies Taylor expansion terms to make the historical features cached from the previous full computation approach true feature values during current reuse. The definition of the $i$-th forward finite difference is:

$$\Delta^i \mathcal{F}(x_t) = \Delta(\Delta^{i-1}\mathcal{F}(x_t)) = \Delta^{i-1}\mathcal{F}(x_{t+N}) - \Delta^{i-1}\mathcal{F}(x_t) \tag{4}$$

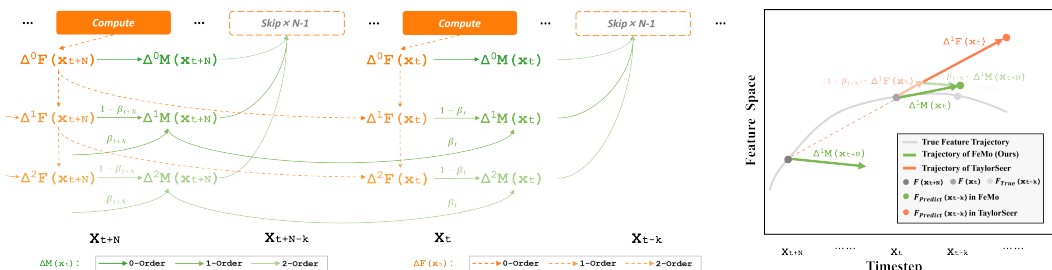

Figure 4: (a) Left: FeMo (Order=2) uses first-order and second-order finite difference approximation derivatives, modeling and predicting the current reuse step feature by utilizing the historical information of each respective order. (b) Right: From a conceptual perspective, the high-dimensional features are abstracted into a 2D space for vector analysis, illustrating that FEMO, when introducing the momentum term for prediction modeling, can make more accurate predictions in both direction and magnitude by considering the historical information of all previous full activation steps.

Where $t$ is the current time step, and $t + N$ is the previous full computation step. And setting the base case $\Delta^0 \mathcal{F}(x_t) = \mathcal{F}(x_t)$. Substituting Eq. 4 into the standard Taylor expansion, the general expression for approximating the reused features using the full computation step features is obtained:

$$\mathcal{F}(x_{t-k}) = \mathcal{F}(x_t) + \sum_{i=1}^{m} \frac{\Delta^i \mathcal{F}(x_t)}{i! \cdot N^i}(-k)^i \tag{5}$$

Although this method can effectively improve the accuracy under feature reuse, using only the expansion terms of the current full activation step as the direction guidance still lacks precision in determining the prediction direction in the vector space. This prediction can even have a negative effect at initial timesteps where feature changes are more significant. This still limits the model from achieving a larger speedup while maintaining generation quality. Therefore, we propose using the **historical gradient** to guide the prediction direction of the current timestep.

## 3.2 FEMO

**Feature prediction during the reuse step.** In order to suppress the oscillations caused by advancing in the direction of the finite difference approximation of the derivative, FEMO introduces a weighted historical momentum term based on the finite difference approximation. This helps smooth out short-term fluctuations in the prediction curve and further correct the predicted direction during the cache reuse step. The iterative formula for the momentum term that stores the historical difference terms is:

$$\mathcal{M}^i(x_t) = \beta \cdot \mathcal{M}^i(x_{t+N}) + (1 - \beta) \cdot \mathcal{F}^{(i)}(x_t) \tag{6}$$

Here, $\beta$ represents the weight of historical information in each iteration, and $(1 - \beta)$ represents the weight of the differential derivative term calculated from the current full activation timestep. The relationship for approximating the derivative using:

$$\Delta^i \mathcal{F}(x_t) \approx N^i \cdot \mathcal{F}^{(i)}(x_t) \tag{7}$$

By replacing the difference approximation term in Eq. 5 with the historical momentum term, and considering the proportional factor in Eq. 7, we derive the feature prediction formula for the $k$-th reuse timestep $t$ using the $m$-th order derivative term:

$$\mathcal{F}(x_{t-k}) = \mathcal{M}(x_t) + \sum_{i=1}^{m} \frac{\mathcal{M}^i(x_t)}{i!}(-k)^i \tag{8}$$

Here, $i$ is the order of differentiation, and $\mathcal{M}(x_t) = \mathcal{F}(x_t)$. At this point, the feature prediction direction at timestep $t$ during cache reuse step is determined not only by the finite difference approximation of the derivative at the previous full activation step, but also by the predicted direction $\mathcal{M}^i(x_{t+N})$, obtained through a weighted vector average of all previously accumulated activation steps in the vector space. The optimization direction tends to adjust gradually along the previous direction, reflecting inertia and directional tendency in the optimization process.

**Weighted Prediction Mechanism.** As shown in Eq. 6, FEMO is a method that assigns different weights to all previous full computation steps, calculates the weighted moving average of local features, and uses the final moving average as the basis for determining the predicted feature values during the reuse step. To understand the weight of the influence of previous full computation steps on the current predicted features, and to make a reasonable initial setting, we derive the following general formula(We perform it when $m = 1$):

$$\mathcal{M}(x_t) = \beta^\tau \cdot \mathcal{M}(T) + (1 - \beta) \cdot (\frac{\mathcal{F}(x_t)}{N} - \beta^{\tau-1} \cdot \frac{\mathcal{F}(x_{t+\tau N})}{N}) - \sum_{j=1}^{\tau-1} \beta^{j-1} \cdot (1-\beta)^2 \cdot \frac{\mathcal{F}(x_{t+jN})}{N}$$

(9)

Here, $t = T\%N$, $\tau = \frac{T-t}{N}$ and $T$ is the full activation step closest to the first feature reuse step. It typically does not directly equal the total number of timesteps.As observed from the Eq. 9, in the early stages of inference, the prediction direction and accuracy mainly depend on the computed values of $\mathcal{M}(T)$ and $\mathcal{F}(x_t)$. Due to the lack of sufficient historical data, the model's prediction error may be large. Given that generative models typically rely on limited prior information and data, we start saving the finite difference approximation values from the first timestep. This way, a larger initial value setting can provide stronger guidance in the early stages of the generation process, making it closer to the target distribution and reducing early fluctuations.

**Bias Correction.** At the same time, as observed in Figure 2 in the introduction, the feature trajectory is relatively smooth during the first few timesteps, with the finite difference derivative values being small (especially in higher-order approximations). To enhance the numerical stability of the FEMO method and ensures more accurate predictions, we have applied bias correction:

$$b = 1 - \beta^{\Delta t}, \mathcal{M}^i(x_t) = \frac{\mathcal{M}^i(x_t)}{b},$$

(10)

where $b$ is the bias correction term, and $\Delta t$ represents the number of timesteps in the current sampling process. When $\Delta t$ is close to 0, the denominator can effectively amplify the current feature value, while when $\Delta t$ is close to $T$, it has almost no impact on the current feature in practice.

### 3.3 ADAPTED-FEMO

At the same time, we observe that when the feature trajectory is relatively smooth, assigning smaller weights to the historical gradients is sufficient for accurate predictions. However, in cases where the trajectory is not smooth—that is, when the values in the cache differ significantly from the true computed values—we can adaptively adjust the momentum term's weight based on the size of the local values. This allows us to effectively determine the prediction direction when significant changes occur in the feature's direction in the vector space. Therefore, we propose that FEMO perform an additional computation step during prediction, i.e., when $t$ in Eq. 9 corresponds to the full computation step: (Mathematical analysis using $m = 1$ as an example).

$$\begin{aligned}
\hat{y} &= \mathcal{M}^0(x_{t+N}) + N \cdot \mathcal{M}^1(x_{t+N}) \\
&= N[\cdot\beta^\tau \cdot \mathcal{M}(T) + (1-\beta) \cdot \mathcal{F}(x_t)] - N \cdot \beta^{\tau-1} \cdot \mathcal{F}(x_{t+\tau N}) + \mathcal{F}(x_{t+N}) \\
&\quad - \sum_{j=1}^{\tau-1} \beta^{j-1} \cdot (1-\beta)^2 \cdot \mathcal{F}(x_{t+jN})
\end{aligned}$$

(11)

At this point, we assume the objective is to minimize the mean squared error between the predicted value $formula\_value$(denoted as $\hat{y}$) and the computed value $true\_value$(denoted as $y$):

$$\min \quad \|y - \hat{y}\|_2$$

(12)

At the same time, we solve the constraint function, so that at the current step $t$,all terms in the objective function, except for the variable $\beta$, are known tensors. Through first-order derivative analysis (Theorem A.2 in the appendix), we can deduce that $\beta$ should satisfy:

$$\beta = \frac{(1-\tau) \cdot \mathcal{F}(x_{t+N})}{\tau \cdot N \cdot \mathcal{M}(T) - \mathcal{F}(x_{t+N})}$$

(13)

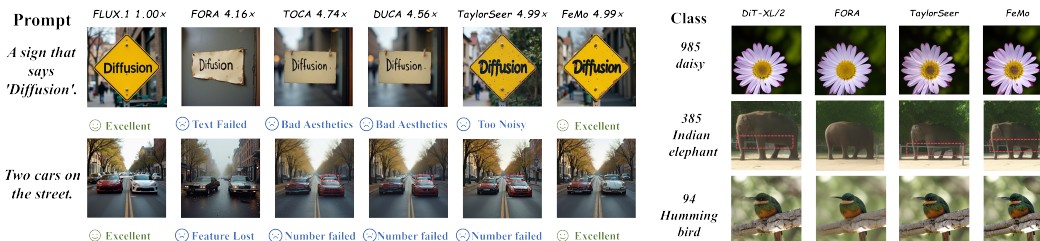

Figure 5: Qualitative comparison on FLUX.1-dev. Other methods encounter issues such as failure in text, wrong number of objects, low aesthetics, and so on, while FEMO achieves the best quality and acceleration.

Figure 6: Detailed visualization results of different acceleration methods on DiT-XL/2 in the case of speed ratio of $6.22\times$.

When $\|y\|_2 > \|\hat{y}\|_2$, it is clear that $\beta$ should tend to increase and change relatively slowly. In order to achieve the optimization objective with the minimal computational effort, we set the learning rate $\gamma$ and determine its specific value based on experimental experience. To implement the adaptive $\beta$, we use the following formula $\beta_t = \beta_{t+N} + \mathbf{S} \cdot \gamma$. When $\|\hat{y}\|_2 - \|y\|_2 < 0$, $\mathbf{S}$ is 1; otherwise, $\mathbf{S}$ is -1, if the difference is exactly 0, $\mathbf{S}$ is set to 0. At the same time, based on Eq. 13 and using a small sample experiment, we derive the initial value for $\beta$ and restrict its range of variation.

### 3.4 ERROR BOUNDS ANALYSIS

We derive the error bound of FEMO. Here, $\mathcal{F}$ denotes the feature function, while FeMo introduces the momentum term $\mathcal{M}$, whose initialization and decay properties ensure faster convergence of higher-order terms. The error bound of FeMo is given by:

$$E_m^{FeMo}(k) \leq \frac{(1 - |\beta|)\sup_{\xi \in [t-k,t]} \|\Delta^m \mathcal{F}(x_\xi)\|}{(m+1)!N^{m+1}}|k|^{m+1} + \sum_{i=1}^{m} \frac{C_i}{i!}|k|^i|N|^{i-1} \quad (14)$$

and it satisfies $E_m^{FeMo}(k) \leq E_m^{TaylorSeer}(k)$. This clearly demonstrates that FEMO consistently achieves a provably tighter theoretical error bound and generally requires only about half the maximum order of TaylorSeer to reliably reach comparable overall performance.

## 4 EXPERIMENT

### 4.1 EXPERIMENT SETTINGS

**Model Configurations.** The experiments are carried out using three advanced visual generative models for our experiments in this study: FLUX.1-dev Labs (2024), a powerful text-to-image generation model, and DiT-XL/2 Peebles & Xie (2023), a widely adopted class-conditional image generation model. For more detailed model configurations, please refer to the Supplementary Material. *FLUX.1-dev* utilizes the Rectified Flow Liu et al. (2023) sampling method with a standard configuration of 50 steps by default. All experimental evaluations were conducted on NVIDIA H20-NVLink GPUs. *DiT-XL/2* adopts a 50-step DDIM Song et al. (2021) sampling strategy to ensure consistency with other models. Experiments on DiT-XL/2 were conducted on NVIDIA A800 80GB GPUs.

**Evaluation and Metrics.** In the text-to-image generation task, we performed inference on 200 prompts from **DrawBench** Saharia et al. (2024) to generate images with a resolution of 1000x1000, using **Image Reward** Xu et al. (2023) and **CLIP score** Hessel et al. (2022) as the primary evaluation metrics. For the class-conditioned image generation task, we uniformly sampled from 1,000 **ImageNet** Russakovsky et al. (2015) categories and generated 50,000 images with a resolution of 256x256, using **FID-50k** Heusel et al. (2018) as the evaluation criterion, supplemented by **sFID** (Stabilized FID) for robustness evaluation. A detailed description can be found in the appendix.

### 4.2 TEXT-TO-IMAGE GENERATION

**Quantitative Study.** We compared Adapted-FEMO with existing methods. As shown in Table 1 ,although DuCa Zou et al. (2024b) ($\mathcal{N} = 5$) achieves $3.45\times$ FLOPs acceleration with an Image

Table 1: Quantitative comparison in text-to-image generation for FLUX on Image Reward.

| Method FLUX.1Labs (2024) | Efficient Attention Dao et al. (2022) | Acceleration Latency(s)↓ | Speed↑ | FLOPs(T)↓ | Speed↑ | Image Reward↑ DrawBench | CLIP↑ Score |
|---|---|---|---|---|---|---|---|
| [dev]: 50 steps | ✔ | 17.20 | 1.00× | 3719.50 | 1.00× | 0.9898 | 19.604 |
| 60% steps | ✔ | 10.49 | 1.64× | 2231.70 | 1.67× | 0.9739 | 19.526 |
| Δ-DiT ($\mathcal{N}=2$) † | ✔ | 11.87 | 1.45× | 2480.01 | 1.50× | 0.9316 | 19.350 |
| DBcache vipshop.com (2025) | ✔ | 11.42 | 1.51× | 2384.29 | 1.56× | 1.0069 | 19.084 |
| 50% steps † | ✔ | 8.80 | 1.95× | 1859.75 | 2.00× | 0.9429 | 19.325 |
| 40% steps † | ✔ | 7.11 | 2.42× | 1487.80 | 2.62× | 0.9317 | 19.027 |
| 34% steps † | ✔ | 6.09 | 2.82× | 1264.63 | 3.13× | 0.9346 | 18.904 |
| Δ-DiT ($\mathcal{N}=3$) † | ✔ | 8.81 | 1.95× | 1686.76 | 2.21× | 0.8561 | 18.833 |
| Chipmunk Silveria et al. (2025) | ✔ | 8.86 | 1.94× | 1505.87 | 2.47× | 0.9936 | 19.441 |
| FORA ($\mathcal{N}=3$) † Selvaraju et al. (2024) | ✔ | 7.08 | 2.43× | 1320.07 | 2.82× | 0.9227 | 18.950 |
| ToCa ($\mathcal{N}=5$) Zou et al. (2024a) | ✘ | 10.80 | 1.59× | 1126.76 | 3.30× | 0.9731 | 19.030 |
| DuCa ($\mathcal{N}=5$) Zou et al. (2024b) | ✔ | 5.88 | 2.93× | 1078.34 | 3.45× | 0.9896 | 19.595 |
| TaylorSeer ($\mathcal{N}=4, \mathcal{O}=2$) Liu et al. (2025) | ✔ | 6.81 | 2.53× | 1042.28 | 3.57× | 1.0024 | 19.402 |
| Adapted-FeMo ($\mathcal{N}=4, \mathcal{O}=1$) | ✔ | 6.67 | 2.58× | 1042.28 | **3.57×** | **1.0375** | **19.618** |
| FORA ($\mathcal{N}=5$) † Selvaraju et al. (2024) | ✔ | 5.17 | 3.33× | 893.54 | 4.16× | 0.8235 | 18.280 |
| TeaCache ($l=0.8$) † (Liu et al., 2024) | ✔ | 4.98 | 3.58× | 892.35 | 4.17× | 0.8683 | 18.500 |
| ToCa ($\mathcal{N}=8$) † Zou et al. (2024a) | ✘ | 8.47 | 2.03× | 784.54 | 4.74× | 0.9086 | 18.380 |
| DuCa ($\mathcal{N}=6$) † Zou et al. (2024b) | ✔ | 4.89 | 3.52× | 816.55 | 4.56× | 0.9470 | 19.082 |
| TaylorSeer ($\mathcal{N}=6, \mathcal{O}=2$) Liu et al. (2025) | ✔ | 5.19 | 3.31× | 744.81 | **4.99×** | 0.9953 | **19.637** |
| Adapted-FeMo ($\mathcal{N}=6, \mathcal{O}=1$) | ✔ | 5.07 | 3.39× | 744.81 | **4.99×** | **0.9984** | 19.597 |
| FORA ($\mathcal{N}=7$) † Selvaraju et al. (2024) | ✔ | 4.22 | 4.08× | 670.44 | 5.55× | 0.7398 | 17.609 |
| ToCa ($\mathcal{N}=10$) † Zou et al. (2024a) | ✘ | 7.93 | 2.17× | 714.66 | 5.20× | 0.8390 | 18.165 |
| DuCa ($\mathcal{N}=9$) † Zou et al. (2024b) | ✔ | 7.28 | 2.36× | 690.26 | 5.39× | 0.8601 | 18.534 |
| TaylorSeer ($\mathcal{N}=7, \mathcal{O}=2$) Liu et al. (2025) | ✔ | 4.88 | 3.52× | 670.44 | **5.55×** | 0.9331 | 19.553 |
| TeaCache ($l=1.2$) † (Liu et al., 2024) | ✔ | 3.98 | 4.48× | 669.27 | 5.56× | 0.7351 | 18.080 |
| Adapted-FeMo ($\mathcal{N}=7, \mathcal{O}=1$) | ✔ | 4.70 | 3.66× | 670.44 | **5.55×** | **0.9770** | **19.556** |
| FORA ($\mathcal{N}=9$) † Selvaraju et al. (2024) | ✔ | 4.42 | 3.90× | 596.07 | 6.24× | 0.5550 | 18.371 |
| ToCa ($\mathcal{N}=12$) † Zou et al. (2024a) | ✘ | 7.34 | 2.34× | 644.70 | 5.77× | 0.7131 | 17.907 |
| DuCa ($\mathcal{N}=10$) † Zou et al. (2024b) | ✔ | 6.5 | 2.65× | 606.91 | 6.13× | 0.8396 | 18.534 |
| TeaCache ($l=1.4$) † (Liu et al., 2024) | ✔ | 3.63 | 4.91× | 594.90 | 6.25× | 0.7346 | 17.862 |
| TaylorSeer ($\mathcal{N}=8, \mathcal{O}=2$) †Liu et al. (2025) | ✔ | 4.59 | 3.74× | 596.07 | **6.24×** | 0.8167 | 19.499 |
| Adapted-FeMo ($\mathcal{N}=8, \mathcal{O}=1$) | ✔ | 4.37 | 3.94× | 596.07 | **6.24×** | **0.9501** | **19.550** |

- † Methods exhibit significant degradation in Image Reward, leading to severe deterioration in image quality.

Table 2: Comparison experiment between FeMo and the baseline on U-Net based SDXL.

| Method | ImageReward↑ | LPIPS↓ | Speed | Latency |
|---|---|---|---|---|
| SD-XL | 0.4535 | 0.000 | 1.00× | 1.038s |
| Deepcache ($\mathcal{N}=2$) | 0.4455 | **0.133** | 1.34× | 0.774s |
| Taylorseer ($\mathcal{N}=2, \mathcal{O}=1$) | 0.4918 | 0.276 | 1.50× | 0.691s |
| FeMo ($\mathcal{N}=2, \mathcal{O}=1$) | **0.4987** | 0.264 | 1.48× | 0.700s |
| Deepcache ($\mathcal{N}=7$) | -2.2052 | 0.845 | 1.71× | 0.606s |
| Taylorseer ($\mathcal{N}=7, \mathcal{O}=1$) | 0.3777 | 0.465 | 2.19× | 0.473s |
| FeMo ($\mathcal{N}=7, \mathcal{O}=1$) | **0.4340** | **0.433** | 2.16× | 0.479s |

Table 3: Comparison of different methods on FID and LPIPS on FLUX.

| Method | FID↓ | LPIPS↓ | FLOPS | Speed |
|---|---|---|---|---|
| DuCa ($\mathcal{N}=9$) | 44.55 | 0.552 | 690.26 | 5.39× |
| Teacache ($l=1.2$) | 34.88 | 0.671 | 669.27 | 5.56× |
| FORA ($\mathcal{N}=7$) | 34.79 | 0.539 | 670.44 | 5.55× |
| ToCa ($\mathcal{N}=10$) | 33.81 | 0.479 | 714.66 | 5.20× |
| TaylorSeer ($\mathcal{N}=7, \mathcal{O}=2$) | 28.31 | 0.452 | 670.44 | 5.55× |
| FeMo ($\mathcal{N}=7, \mathcal{O}=1$) | **25.16** | **0.384** | 670.44 | **5.55×** |

Reward of 0.9896, and ToCa Zou et al. (2024a) ($\mathcal{N}=5$) provides 3.30× acceleration, its image quality noticeably drops (0.9731). However, the overall performance of Adapted-FeMo ($\mathcal{N}=4$, $\mathcal{O}=1$) significantly outperforms both: with **3.57× acceleration**, it consistently maintains an excellent Image Reward of **1.0375**. In comparison to the recent cache-based high-acceleration method TaylorSeer Liu et al. (2025), which retains a stable Image Reward of 0.9331 at 5.55× acceleration, our Adapted-FeMo maintains an even better Image Reward (0.9770) and CLIP score (19.556) at the same **5.55× acceleration**. Notably, as the acceleration ratio increases, baseline methods suffer a significant progressive degradation in image quality: ToCa ($\mathcal{N}=12$) drops to 0.7131 Image Reward at 5.77× acceleration, DuCa ($\mathcal{N}=10$) drops to 0.8396 Image Reward at 6.13× acceleration, and TaylorSeer ($\mathcal{N}=8$, $\mathcal{O}=2$) drops to 0.8167 Image Reward at 6.24× acceleration. In contrast, Adapted-FeMo ($\mathcal{N}=8$, $\mathcal{O}=1$) maintains an Image Reward of 0.9501 and a CLIP score of 19.550 even at **6.24× acceleration**, demonstrating an unparalleled balance of efficiency and fidelity.

**Qualitative Study.** Qualitative results in Figure 5 clearly demonstrate that FeMo achieves outstanding generation quality while still enabling high-speed inference. Here, *Feature Lost* refers to the absence of key information contained in the prompt compared to the original image. In the text generation task, such as *A sign that says 'Diffusion'*, FeMo accurately preserves all the textual elements, whereas methods like **ToCa** and **DuCa** often lose key details. In the generation task *Two cars on the street*, FeMo exhibits a strong semantic ability to fully understand the prompt, while other methods show significant issues with both color accuracy and quantity accuracy in the test cases. This indicates that FeMo strikes an excellent balance between speed and performance, especially in tasks that require fine detail preservation and a strong understanding of the prompt.

Table 4: Quantitative comparison on class-to-image generation on ImageNet with DiT-XL/2.

| Method | Efficient | Acceleration | | | FID ↓ | sFID ↓ |
|---|---|---|---|---|---|---|
| DiT-XL/2 Peebles & Xie (2023) | Attention Dao et al. (2022) | Latency(s) ↓ | FLOPs(T) ↓ | Speed ↑ | | |
| **DDIM-50 steps** | ✔ | 0.505 | 23.74 | 1.00× | 2.32 | 4.32 |
| **DDIM-25 steps** | ✔ | 0.273 | 11.87 | 2.00× | 2.95 | 4.51 |
| **Δ-DiT**($\mathcal{N}=2$) | ✔ | 0.322 | 18.04 | 1.31× | 2.69 | 4.67 |
| **Δ-DiT**($\mathcal{N}=3$) | ✔ | 0.301 | 16.14 | 1.47× | 3.75 | 5.70 |
| **DDIM-20 steps** | ✔ | 0.215 | 9.49 | 2.50× | 3.81 | 5.15 |
| **FORA** ($\mathcal{N}=3$) Selvaraju et al. (2024) | ✔ | 0.197 | 8.58 | 2.77× | 3.55 | 6.36 |
| **ToCa** ($\mathcal{N}=3$) Zou et al. (2024a) | ✘ | 0.216 | 10.23 | 2.32× | 2.87 | 4.76 |
| **DuCa** ($\mathcal{N}=3$) Zou et al. (2024b) | ✔ | 0.208 | 9.58 | 2.48× | 2.88 | 4.66 |
| **TaylorSeer** ($\mathcal{N}=3, \mathcal{O}=4$) Liu et al. (2025) | ✔ | 0.292 | 8.56 | 2.77× | 2.35 | 4.69 |
| **Adapted-FEMO** ($\mathcal{N}=3, \mathcal{O}=2$) | ✔ | 0.241 | 8.56 | 2.77× | **2.32** | **4.65** |
| **DDIM-12 steps**† | ✔ | 0.141 | 5.70 | 4.17× | 7.80 | 8.03 |
| **FORA** ($\mathcal{N}=4$) Selvaraju et al. (2024) | ✔ | 0.169 | 6.66 | 3.56× | 4.30 | 7.37 |
| **ToCa** ($\mathcal{N}=6$)†Zou et al. (2024a) | ✘ | 0.163 | 6.34 | 3.75× | 6.55 | 7.10 |
| **DuCa** ($\mathcal{N}=6$)† Zou et al. (2024b) | ✔ | 0.127 | 5.81 | 4.08× | 6.40 | 6.71 |
| **TaylorSeer** ($\mathcal{N}=5, \mathcal{O}=4$)Liu et al. (2025) | ✔ | 0.245 | 5.24 | 4.53× | 2.74 | 5.82 |
| **Adapted-FEMO** (N = 5, O = 2) | ✔ | 0.166 | 5.24 | **4.53×** | **2.64** | **5.30** |
| **DDIM-10 steps**† | ✔ | 0.126 | 4.75 | 5.00× | 12.15 | 11.33 |
| **FORA** ($\mathcal{N}=7$)† Selvaraju et al. (2024) | ✔ | 0.142 | 3.82 | 6.22× | 12.55 | 18.63 |
| **ToCa** ($\mathcal{N}=13$)† Zou et al. (2024a) | ✘ | 0.146 | 4.03 | 5.90× | 21.24 | 19.93 |
| **DuCa** ($\mathcal{N}=12$)† Zou et al. (2024b) | ✔ | 0.131 | 3.94 | 6.02× | 31.97 | 27.26 |
| **TaylorSeer** (N = 7, O = 4)Liu et al. (2025) | ✔ | 0.220 | 3.82 | 6.22× | 3.59 | 7.07 |
| **Adapted-FEMO** ($\mathcal{N}=7, \mathcal{O}=2$) | ✔ | 0.133 | 3.82 | **6.22×** | **3.36** | **5.63** |
| **DDIM-7 steps**† | ✔ | 0.095 | 3.32 | 7.14× | 33.65 | 27.15 |
| **FORA** ($\mathcal{N}=8$)† Selvaraju et al. (2024) | ✔ | 0.141 | 3.34 | 7.10× | 15.31 | 21.91 |
| **ToCa** ($\mathcal{N}=13$)† Zou et al. (2024a) | ✘ | 0.151 | 3.66 | 6.48× | 22.18 | 20.68 |
| **DuCa** ($\mathcal{N}=18$)† Zou et al. (2024b) | ✔ | 0.144 | 3.59 | 6.61× | 133.06 | 98.13 |
| **TaylorSeer** ($\mathcal{N}=9, \mathcal{O}=4$)†Liu et al. (2025) | ✔ | 0.209 | 3.34 | 7.10× | 5.55 | 8.45 |
| **Adapted-FEMO** (N = 9, O = 2) | ✔ | 0.122 | 3.34 | **7.10×** | **4.46** | **5.99** |

- † Methods exhibit significant degradation in FID, leading to severe deterioration in image quality.

### 4.3 CLASS-CONDITIONAL IMAGE GENERATION

**Quantitative Study.** We compared **Adapted-FEMO** with ToCa Zou et al. (2024a), FORA Selvaraju et al. (2024), DuCa Zou et al. (2024b), TaylorSeer Liu et al. (2025), and methods that reduce DDIM steps on DiT-XL/2 Peebles & Xie (2023). The results show that Adapted-FEMO significantly outperforms other methods in terms of both acceleration ratio and image quality. As the acceleration ratio increases beyond 3.5×, the FID scores of methods like FORA, ToCa, and DuCa degrade significantly, leading to severe deterioration in image quality. In contrast, Adapted-FEMO maintains excellent generation quality even at **4.53× acceleration**, with a **FID of 2.68** and sFID of 5.30,superior to advanced baselines such as TaylorSeer, ToCa, and DuCa. Notably, Adapted-FEMO can still maintain good generation quality, without image degradation, even at the highest acceleration of **7.10×**, achieving an outstanding balance between efficiency and fidelity.

**Qualitative Study.** The qualitative results in Figure 6 demonstrate that FEMO successfully maintains both the details and quality of the images during high-speed inference on the DiT-XL/2 model. In the generation task for the "*985 daisy*" class, FEMO accurately preserved the subtle structural details of the flower. In the "*385 Indian elephant*" generation task, FEMO successfully modeled the relationship between the elephant's legs and the position of the fence, showing a good understanding of the physical spatial details and overall generation capabilities, in contrast to the baseline FORA, which failed to generate the outline, and TaylorSeer, which lacked important modeling details.

### 4.4 FURTHER EXPERIMENTAL ANALYSIS

Table 2 reports additional comprehensive comparisons between FEMO and baseline methods on a representative SD-XL model with a U-Net backbone. Even under a high acceleration setting ($\mathcal{N}=7$), FEMO still maintains stable generation quality while achieving superior practical speedup over the baselines. Table 3 further evaluates FEMO on FLUX across several different benchmarks. The results show that the choice of benchmark has little observable impact on the performance of FEMO which consistently preserves its advantages in both image quality and sampling efficiency.

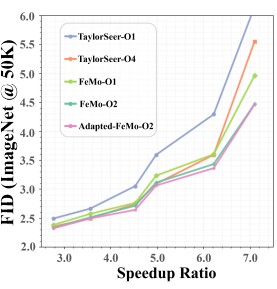 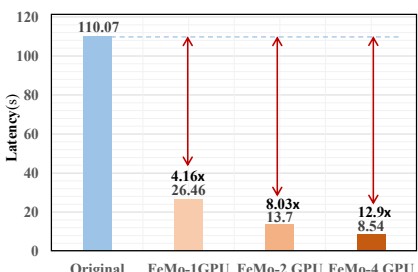

Figure 8: Comparison between baselines and FEMO on DrawBench with FLUX and ImageNet with DiT.

Figure 9: Scatter plot of the trajectories of FEMO and baselines after PCA.

# 5 DISCUSSION

## 5.1 ABLATION STUDIES

We conducted ablation experiments on DiT-XL/2 Peebles & Xie (2023) and FLUX.1-devLabs (2024) to evaluate Adapted-FEMO and FEMO, focusing on the impact of the interval time parameter $\mathcal{N}$ and the order of the differential approximation $\mathcal{O}$ on computational efficiency and generation quality. The results show that, when using a first-order differential approximation ($\mathcal{O} = 1$), Adapted-FEMO significantly outperforms the current state-of-the-art TaylorSeer on FLUX. On DiT, at a high acceleration ratio ($\mathcal{N} = 9$), it also surpasses TaylorSeer in its full state ($\mathcal{O} = 4$). Furthermore, when $\mathcal{O} = 2$, Adapted-FEMO shows a noticeable improvement on DiT. At $\mathcal{N} = 7$, it still achieves an FID of only 3.36 and can maintain image generation quality without degradation at $\mathcal{N} = 9$. Meanwhile, the ablation results on FEMO show that our adaptive adjustment strategy continues to improve performance without affecting generation speed. The ablation experiments also clearly demonstrate that using detailed differential approximation derivative information from multiple historical time steps can effectively and consistently enhance overall prediction quality.

Overall, the Adapted-FEMO method demonstrates significant advantages in the current ablation experiments, especially in its performance at high acceleration ratios. Compared to existing methods, Adapted-FEMO achieves a higher acceleration ratio, a breakthrough that lays the foundation for its application in real-time or resource-constrained scenarios. Detailed results can be found in the C.

## 5.2 THE STABILITY OF ITS HYPERPARAMETERS.

We chose to analyze the Adapted-FEMO ($\mathcal{N}$=9, $\mathcal{O}$=2) scheme on DiT-XL/2, where different $\gamma$ values were analyzed within the same fluctuation range, as well as different ranges for the same $\gamma$, demonstrating the stability of hyperparameters within reasonable ranges, with maximum fluctuations of only 0.16% and 0.19%, respectively. More analysis can be found in the D.

## 5.3 FEMO IN SEQUENCE PARALLELISM TECHNOLOGY.

As shown in the Figure 9, FEMO is highly compatible with sequence parallelism technology. When generating images with a resolution of 2048, the latency on a single GPU is reduced from 26.46 to 13.70, achieving a 1.93× speedup. On four GPUs in parallel, the latency is reduced from 26.46 to 8.54, achieving a 3.10× speedup, indicating compatibility with parallel computation.

# 6 CONCLUSION

In this paper, to address the existing issues in the "cache-then-forecast" paradigm—where current methods are highly sensitive to gradient accumulation influenced by noise, struggle to handle long-term dependencies, and often overlook the subtle feature trajectory differences between different generated samples—we propose the FEMO method based on a weighted prediction mechanism for improved stability. This method uses the differential approximation of derivatives from previously fully activated timesteps to predict the features at the current reuse step in a more principled manner. Additionally, we introduce an adaptive mechanism that dynamically adjusts the weight of historical features during momentum updates based on each sample's feature trajectory characteristics.

## ETHICS STATEMENT

This work adheres to the ICLR Code of Ethics. Our research is conducted entirely on publicly available datasets and does not involve any personally identifiable or sensitive information. The proposed method is intended solely for academic research purposes. This study does not introduce new ethical risks beyond those inherent to underlying diffusion models. In our experiments, we only employ publicly available models and datasets, and our acceleration technique is model-agnostic and content-neutral. While our method reduces inference time and computational cost, potentially making generative AI more accessible, we acknowledge that such accessibility applies both to beneficial use cases and to potentially harmful ones. We encourage responsible deployment of accelerated diffusion models in accordance with existing ethical guidelines for AI-generated content, including appropriate disclosure of synthetic media and consideration of potential societal impacts.

## REPRODUCIBILITY STATEMENT

We are committed to ensuring the full reproducibility of our FEMO framework. To this end, Section 3 provides the complete mathematical formulations of the core algorithmic components. All experimental configurations are detailed in Section 4.1 and the Appendix, including the models evaluated (e.g., FLUX-1-dev, DiT-XL/2), the datasets used (DrawBench and ImageNet), and the full set of evaluation metrics (e.g., ImageReward, CLIP Score). The Appendix further presents our detailed ablation studies and hyperparameter choices for decomposition methods and prediction strategies. Source code files are provided in the supplementary materials and will be released in a public repository upon acceptance.

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

# Supplementary Material

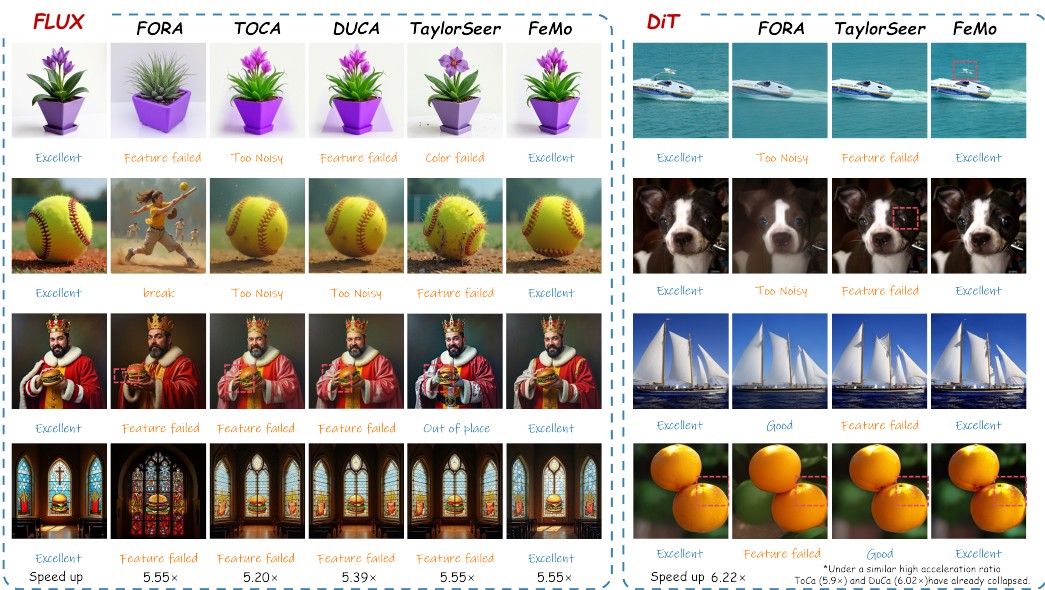

Figure 10: **A gallery comparing various acceleration methods and generation quality**. We introduce FEMO, which observes the continuity of feature trajectory of diffusion models in different timesteps and further stabilizes it with an adaptive *momentum* mechanism, leading to $5.55\times$ and $6.22\times$ acceleration in FLUX and DiT without notable drop in generation quality.

## A  CONCLUSION PROOF

### A.1  PROOF OF EQ. 9

**Theorem A.1** (Iterative formula for historical momentum term). *Let $i$ be the current order of the finite difference approximation derivative, $m$ be the maximum order, and $\beta$ be the weight of the momentum term. Therefore, the weight of the current finite difference approximation derivative is $(1-\beta)$. Here, $t$ represents the current timestep, and $N$ is the distance between two consecutive full computation steps. Based on Eq. 6, for any $i$ in the range $(0, m)$, the following general formula represents the weight relationship between previous full computation steps and the current predicted feature:*

$$\mathcal{M}(x_t) = \beta^\tau \cdot \mathcal{M}(T) + (1-\beta) \cdot \left(\frac{\mathcal{F}(x_t)}{N} - \beta^{\tau-1} \cdot \frac{\mathcal{F}(x_{t+\tau N})}{N}\right)$$

$$- \sum_{j=1}^{\tau-1} \beta^{j-1} \cdot (1-\beta)^2 \cdot \frac{\mathcal{F}(x_{t+jN})}{N}$$

*Proof.* In this context, we use $\mathcal{F}^i(x_t)$ to represent $\Delta^i \mathcal{F}(x_t)$, which denotes the $i$-th order partial derivative of $\mathcal{F}(x_t)$ with respect to $x_t$.

$$\mathcal{M}^i(x_t) = \beta \cdot \mathcal{M}^i(x_{t+N}) + (1 - \beta) \cdot \mathcal{F}^{(i)}(x_t)$$

$$= \beta \cdot \mathcal{M}^i(x_{t+N}) + (1 - \beta) \cdot \frac{\mathcal{F}^{i-1}(x_t) - \mathcal{F}^{i-1}(x_{t+N})}{N^i}$$

$$= \beta^2 \cdot \mathcal{M}^i(x_{t+2N}) + \beta \cdot (1 - \beta) \cdot \frac{\mathcal{F}^{i-1}(x_{t+N}) - \mathcal{F}^{i-1}(x_{t+2N})}{N^i} + (1 - \beta) \cdot \frac{\mathcal{F}^{i-1}(x_t) - \mathcal{F}^{i-1}(x_{t+N})}{N^i}$$

$$= \beta^2 \cdot \mathcal{M}^i(x_{t+2N}) + (1 - \beta) \cdot \frac{\mathcal{F}^{i-1}(x_t)}{N^i} - (1 - \beta)^2 \cdot \frac{\mathcal{F}^{i-1}(x_{t+N})}{N^i} - \beta \cdot (1 - \beta) \cdot \frac{\mathcal{F}^{i-1}(x_{t+2N})}{N^i}$$

$$= \beta^3 \cdot \mathcal{M}^i(x_{t+3N}) + (1 - \beta) \cdot \frac{\mathcal{F}^{i-1}(x_t)}{N^i} - (1 - \beta)^2 \cdot \frac{\mathcal{F}^{i-1}(x_{t+N})}{N^i}$$

$$- \beta \cdot (1 - \beta)^2 \cdot \frac{\mathcal{F}^{i-1}(x_{t+2N})}{N^i} - \beta^2 \cdot (1 - \beta) \cdot \frac{\mathcal{F}^{i-1}(x_{t+3N})}{N^i}$$

$$....$$

$$= \beta^\tau \cdot \mathcal{M}^i(T) + (1 - \beta) \cdot \left( \frac{\mathcal{F}^{i-1}(x_t)}{N^i} - \beta^{\tau-1} \cdot \frac{\mathcal{F}^{i-1}(x_{t+\tau N})}{N^i} \right) - \sum_{j=1}^{\tau-1} \beta^{j-1} \cdot (1 - \beta)^2 \cdot \frac{\mathcal{F}^{i-1}(x_{t+jN})}{N}$$

### A.2 Proof of Eq. 13

**Theorem A.2** (Solution to the constraint function of $\beta$). *Let $i$ be the current order of the finite difference approximation derivative, In this analysis of the theorem, we take $i == 1$. $t = T\%N$, $\tau = \frac{T-t}{N}$ and $T$ is the full computation step closest to the first feature reuse step. We derive the expression for the local extrema of $\beta$ in the LOSS.*

$$\beta = \frac{(1 - \tau) \cdot \mathcal{F}(x_{t+N})}{\tau \cdot N \cdot \mathcal{M}(T) - \mathcal{F}(x_{t+N})}$$

*Finally, in the small sample experiment, we referenced the theoretical extremum points, mainly using experimental verification to determine the initial $\beta$ value. When $\|true\_value\|_2 > \|formula\_value\|_2$, we chose to increase $\beta$ by a fixed step size $\gamma$, and conversely, we decreased $\beta$ when the inequality was reversed.*

*Proof.* In this context, We use $y$ to represent $true\_value$, and $\hat{y}$ is $formula\_value$.

$$\frac{\partial L}{\partial \beta} = 2 \cdot \|\hat{y} - y\| \cdot \frac{\partial \hat{y}}{\partial \beta}$$

*and then:*

$$\frac{\partial \hat{y}}{\partial \beta} = \tau \cdot N \cdot \beta^{\tau-1} \cdot \mathcal{M}(T) - \mathcal{F}(x_{t+N}) - [\beta^{\tau-1} - (\tau - 1) \cdot \beta^{\tau-2}] \cdot \mathcal{F}(x_{t+N})$$

$$- \sum_{j=1}^{\tau-1} [(j - 1) \cdot \beta^{j-2} \cdot (1 - \beta)^2 - 2\beta^{j-1} \cdot (1 - \beta)] \cdot \mathcal{F}(x_{t+jN})$$

Therefore, the first derivative of the $L$ can be represented by the following inequality:

$$\frac{\partial L}{\partial \beta} \leq \left\{ \tau N \beta^{\tau-1} \cdot \mathcal{M}(T) - [\beta^{\tau-1} - (\tau - 1) \cdot \beta^{\tau-2}] \cdot \mathcal{F}(x_{t+N}) \right\} \cdot 2\|y - \hat{y}\|$$

We perform a local extremum analysis of the LOSS function based on this inequality and use the scaled inequality to roughly determine the range of $\beta$. This also provides theoretical reference for the experiment on adaptively adjusting it.

## B Experimental Details

In this section, more details of the experiments are provided.

**Model Configuration**   As described in 4.1, we use two models for different tasks, namely FLUX for text-to-image generation and DiT for class-conditional image generation. This section provides more detailed hyperparameter configuration schemes.

- **FLUX**:The FORA method selects the reuse step $N$ between 3 and 9, with an acceleration ratio similar to FEMO. The ToCa method selects $N$ between 5 and 12, with a 90% cache rate and uses an attention-based token selection method. It employs a non-uniform activation interval, starting with sparse activation and transitioning to dense activation. The DuCa method selects $N$ between 5 and 10, using conservative cache steps for even-numbered timesteps and aggressive steps for odd-numbered ones. The activation intervals and cache rate match those of ToCa. TeaCache selects the optimal caching threshold based on acceleration ratios.

- **DiT**:The FORA method selects the reuse step $N$ between 3 and 8, with an acceleration ratio similar to FEMO. ToCa chooses $N$ between 3 and 13, with a 95% cache rate, using an attention-based token selection method and a non-uniform activation interval, starting with sparse activation and transitioning to dense activation. The DuCa method selects $N$ between 3 and 18, with aggressive cache steps for odd-numbered timesteps. The activation interval and cache rate match those of

All models include a unified forced activation period $N$, where $\beta$ is the momentum coefficient, and the first-order derivative term coefficient is $1 - \beta$. Adaptive algorithm step sizes $\gamma$ and adjustment limits are also used to optimize computational efficiency and model performance. In our experiments, when timestep is start,we assign $\mathcal{M}^0(x_t)$ to $\mathcal{M}^1(x_t)$ in order to achieve dynamic compensation for errors introduced by the derivative approximation via finite differences.

- **FLUX**: The parameter $\beta$ for FEMO is determined by the approximate range of the best selection based on Eq. 13, and the optimal parameter 0.325 is empirically obtained around this theoretical value. The Adapted-FEMO method selects $\gamma$ between 0.01 and 0.015, with slight differences at different acceleration ratios, and the upper and lower bounds for $\beta$ are between 0.2 and 0.45.

- **DiT**:The parameter $\beta$ for FEMO is determined by the approximate range of the best selection based on equation 13, and the optimal parameter is empirically obtained around this theoretical value. When $\mathcal{O} = 1$, $\beta$ is selected as -0.2; when $\mathcal{O} = 2$, $\beta$ is selected in the range of -0.01 to -0.03 as the initial value, with slight differences for different acceleration ratios. The $\gamma$ for Adapted-FEMOis selected between 0.001 and 0.01, with slight variations at different acceleration ratios. Additionally, the upper and lower bounds for $\beta$ are between -0.04 and 0.

## C   SUPPLEMENTARY RESULTS FOR ABLATION STUDIES

The results of the ablation experiments under different configurations are presented in Table 1 to 4.

Table 1: Ablation Study of FEMO with Different Configurations on ImageNet with DiT-XL/2.

| Configuration | FLOPs↓ | Speed↑ | FID↓ | sFID↓ |
|---|---|---|---|---|
| ($\mathcal{N}$=3, $\mathcal{O}$=1) | 8.56 | 2.77× | 2.38 | 4.72 |
| ($\mathcal{N}$=4, $\mathcal{O}$=1) | 6.66 | 3.56× | 2.57 | 5.25 |
| ($\mathcal{N}$=5, $\mathcal{O}$=1) | 5.24 | 4.53× | 2.76 | 5.31 |
| ($\mathcal{N}$=6, $\mathcal{O}$=1) | 4.76 | 4.98× | 3.23 | 6.52 |
| ($\mathcal{N}$=7, $\mathcal{O}$=1) | 3.82 | 6.22× | 4.60 | 6.94 |
| ($\mathcal{N}$=8, $\mathcal{O}$=1) | 3.82 | 6.22× | 4.96 | 8.05 |
| ($\mathcal{N}$=3, $\mathcal{O}$=2) | 8.56 | 2.77× | 2.33 | 4.72 |
| ($\mathcal{N}$=4, $\mathcal{O}$=2) | 6.66 | 3.56× | 2.51 | 5.25 |
| ($\mathcal{N}$=5, $\mathcal{O}$=2) | 5.24 | 4.53× | 2.71 | 5.31 |
| ($\mathcal{N}$=6, $\mathcal{O}$=2) | 4.76 | 4.98× | 3.11 | 6.21 |
| ($\mathcal{N}$=7, $\mathcal{O}$=2) | 3.82 | 6.22× | 3.43 | 6.74 |
| ($\mathcal{N}$=8, $\mathcal{O}$=2) | 3.82 | 6.22× | 4.40 | 7.25 |
| ($\mathcal{N}$=9, $\mathcal{O}$=2) | 3.34 | 7.10× | 4.47 | 5.99 |

Table 2: Ablation Study of Adapted-FEMO with Different Configurations on ImageNet with DiT-XL/2.

| Configuration | FLOPs↓ | Speed↑ | FID↓ | sFID↓ |
|---|---|---|---|---|
| ($\mathcal{N}$=3, $\mathcal{O}$=2) | 8.56 | 2.77× | 2.32 | 4.63 |
| ($\mathcal{N}$=4, $\mathcal{O}$=2) | 6.66 | 3.56× | 2.49 | 5.13 |
| ($\mathcal{N}$=5, $\mathcal{O}$=2) | 5.24 | 4.53× | 2.68 | 5.29 |
| ($\mathcal{N}$=6, $\mathcal{O}$=2) | 4.76 | 4.98× | 3.06 | 6.21 |
| ($\mathcal{N}$=7, $\mathcal{O}$=2) | 3.82 | 6.22× | 3.36 | 5.64 |
| ($\mathcal{N}$=8, $\mathcal{O}$=2) | 3.82 | 6.22× | 4.40 | 6.56 |
| ($\mathcal{N}$=9, $\mathcal{O}$=2) | 3.34 | 7.10× | 4.46 | 5.98 |

Table 3: Ablation Study of FEMO with Different Configurations on DrawBench200 with FLUX.1-dev.

| Configuration | FLOPs↓ | Speed↑ | ImageReward↑ |
|---|---|---|---|
| ($\mathcal{N}$=3, $\mathcal{O}$=1) | 1339.75 | 2.78× | 1.0505 |
| ($\mathcal{N}$=4, $\mathcal{O}$=1) | 1042.28 | 3.57× | 1.0362 |
| ($\mathcal{N}$=5, $\mathcal{O}$=1) | 893.54 | 4.16× | 1.0007 |
| ($\mathcal{N}$=6, $\mathcal{O}$=1) | 744.81 | 4.99× | 0.9950 |
| ($\mathcal{N}$=7, $\mathcal{O}$=1) | 670.44 | 5.55× | 0.9754 |
| ($\mathcal{N}$=8, $\mathcal{O}$=1) | 596.07 | 6.24× | 0.9373 |
| ($\mathcal{N}$=9, $\mathcal{O}$=1) | 596.07 | 6.24× | 0.9157 |
| ($\mathcal{N}$=10, $\mathcal{O}$=1) | 521.71 | 7.13× | 0.8606 |

## D    SUPPLEMENTARY RESULTS FOR 5.2

In this section, we mainly conduct a parameter stability analysis of the Adapted-FEMO (N9O2) scheme on DiT-XL/2. The parameter $\gamma$ is the step size change used in the adaptive update of the historical term weight $\beta$ in the update mechanism of $\mathcal{M}^i(x_t)$.

$\delta$ represents the range within which the adaptive step size change is constrained, with boundary values set to `first_`$\beta$ - $\delta$ and `first_`$\beta$ + $\delta$. When the updated result exceeds this range, it will automatically be assigned to the boundary value. This constraint prevents error accumulation due to inaccurate local optimal $\beta$ calculations when the full activation frequency $N$ is large.

We visualized the FID metric of the generation results of 50,000 samples with $\gamma$ values of 0.001, 0.004, 0.007, and 0.01, and $\delta$ values of 0.01, 0.02, and 0.03. The analysis separately examines the impact of $\gamma$ on generation quality with a fixed $\delta$, and the impact of $\delta$ on generation quality with a fixed $\gamma$. The results show that, with a fixed $\delta$, the impact of $\gamma$ on generation quality is minimal, with a maximum fluctuation difference of only 0.16%.Similarly, with a fixed $\gamma$, the impact of $\delta$ on generation quality is also minimal, with a maximum fluctuation of only 0.19% when $\gamma$ = 0.007. This analysis proves that the adaptive mechanism of Adapted-FEMO is not affected by small numerical fluctuations within reasonable parameter settings and can effectively improve generation quality under high acceleration ratios.

Table 4: Ablation Study of Adapted-FEMO with Different Configurations on DrawBench200 with FLUX.1-dev.

| Configuration | FLOPs↓ | Speed↑ | ImageReward↑ |
|---|---|---|---|
| ($\mathcal{N}$=3, $\mathcal{O}$=1) | 1339.75 | 2.78× | 1.0533 |
| ($\mathcal{N}$=4, $\mathcal{O}$=1) | 1042.28 | 3.57× | 1.0375 |
| ($\mathcal{N}$=5, $\mathcal{O}$=1) | 893.54 | 4.16× | 1.0029 |
| ($\mathcal{N}$=6, $\mathcal{O}$=1) | 744.81 | 4.99× | 0.9984 |
| ($\mathcal{N}$=7, $\mathcal{O}$=1) | 670.44 | 5.55× | 0.9770 |
| ($\mathcal{N}$=8, $\mathcal{O}$=1) | 596.07 | 6.24× | 0.9501 |
| ($\mathcal{N}$=9, $\mathcal{O}$=1) | 596.07 | 6.24× | 0.9235 |
| ($\mathcal{N}$=10, $\mathcal{O}$=1) | 521.71 | 7.13× | 0.8678 |

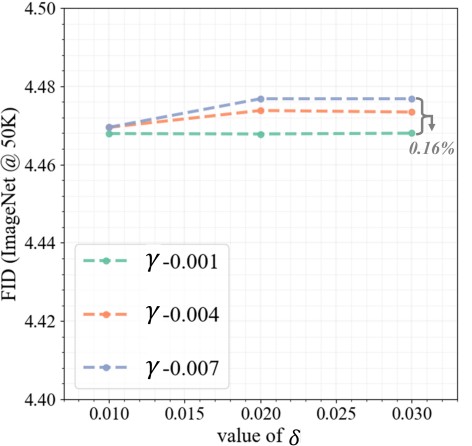

Figure 1: The impact of the $\delta$ parameter on FID under three fixed $\gamma$ values.

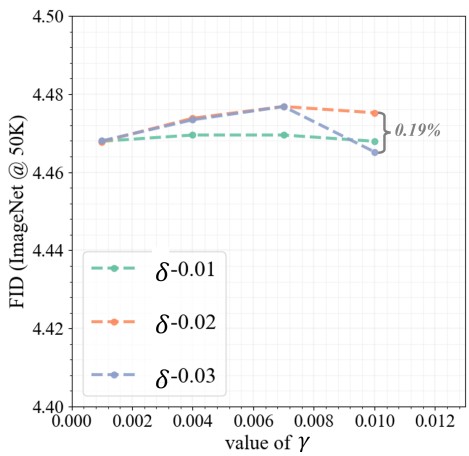

Figure 2: The impact of the $\gamma$ parameter on FID under three fixed $\delta$ values.

## E  ERROR BOUNDS ANALYSIS

We derive the error bound of the proposed **FeMo** and compare it with **TaylorSeer**, showing that FeMo yields smaller prediction errors under identical settings.

For TaylorSeer, the error bound is given by:

$$E_m(k) \leq \frac{M_{m+1}}{(m+1)!}|k|^{m+1}, \quad M_{m+1} = \sup_{\xi \in [t-k,t]} \|\mathcal{F}^{(m+1)}(x_\xi)\|. \tag{15}$$

Here, $\mathcal{F}$ represents the feature function. In contrast, the difference for FeMo lies in replacing $\mathcal{F}$ with the momentum term $\mathcal{M}$:

$$M_{m+1} = \sup_{\xi \in [t-k,t]} \|\mathcal{M}^{(m+1)}(x_\xi)\|.$$

From $\tau = \frac{T-t}{N}$, we can deduce that $\mathcal{F}(x_{t+\tau N}) = \mathcal{F}(X_T)$. Substituting this into Eq. 9 of the main paper, we obtain:

$$M^i(x_t) \leq |\beta|^\tau \mathcal{M}^i(x_T) + (1-|\beta|)\Big(\frac{\mathcal{F}^{i-1}(x_t)}{N^i} - \frac{|\beta|^{\tau-1}\mathcal{F}^{i-1}(x_T)}{N^i}\Big). \tag{16}$$

Based on the initialization settings of FLUX and DiT:

$$M^i(X_T) = \begin{cases} \mathcal{F}(X_T), & i = 0 \text{ or } 1, \\ 0, & \text{otherwise.} \end{cases}$$

Thus, for $i = 1$, we get:

$$\mathcal{M}^1(x_t) < |\beta|^\tau \Big(1 + \frac{|\beta| - 1}{|\beta|N}\Big)\mathcal{F}(x_T) + (1 - |\beta|)\mathcal{F}(x_t). \tag{17}$$

Since $|\beta| \in (0, 1)$, the exponential decay of $\beta^\tau$ ensures that Eq. 2 converges to zero. Moreover, Eq. 3 is always $\leq$ Eq. 15, since Eq. 15 takes the supremum over the interval including Eq. 3. When $i > 1$, $\mathcal{M}^i(X_T) = 0$, and the error term upper bound is given by Eq. 3.

Importantly, FeMo can achieve similar performance using roughly half of the maximum order required by TaylorSeer, which means the differential approximation error in FeMo is significantly smaller.

**Final bound.** The inference error bound for a single sample in FeMo is:

$$E_m^{FeMo}(k) \leq \frac{(1 - |\beta|)\sup_{\xi \in [t-k,t]} \|\Delta^m \mathcal{F}(x_\xi)\|}{(m+1)!N^{m+1}}|k|^{m+1} + \sum_{i=1}^m \frac{C_i}{i!}|k|^i|N|^{i-1}, \tag{18}$$

which satisfies

$$E_m^{FeMo}(k) \leq E_m^{TaylorSeer}(k).$$

Therefore, we theoretically establish the superiority of FeMo over TaylorSeer in terms of error bounds.

## THE USE OF LARGE LANGUAGE MODELS (LLMS)

We only used large language models (LLMs) for polishing certain sentences in the paper to ensure fluency. The key parts of the paper were written by the authors.

