# OpenReview forum: "Accelerate Diffusion Transformers with Feature Momentum"
_ICLR.cc/2026/Conference — Submitted to ICLR 2026_

### Official Review · Reviewer_Vsqw · 2025-10-29

**Soundness:** 3
**Presentation:** 2
**Contribution:** 2
**Rating:** 6
**Confidence:** 4

**Summary:**

This paper proposes a feature momentum mechanism called FeMo, which aims to accelerate diffusion transformers by predicting future step features in the diffusion process using a momentum-inspired adaptive method. Subsequently, Adapted-FeMo is constructed by introducing a momentum coefficient for each sample, further minimizing prediction errors. To verify the effectiveness of FeMo and Adapted-FeMo, the authors conducted extensive comparisons with the latest state-of-the-art baselines in image and text-to-image generation tasks. The results show that while maintaining generation quality, Adapted-FeMo achieves a maximum speedup of 7.10x on the DiT model and a maximum speedup of 6.24x on the FLUX model.

**Strengths:**

1. The experiments are sufficiently comprehensive with rich metrics. This paper conducts benchmark tests on text-to-image and class-conditional tasks, and provides multiple metrics (such as FLOP, latency, FID, sFID, LPIPS, etc.) as well as quantitative performance of various acceleration mechanisms.
2. The paper conforms to the paradigm of "cache first, then predict" and effectively solves the problem of finite difference approximation oscillation existing in this paradigm.
3. The mathematical derivation of the paper is relatively sound, with clear update equations and error proofs for FeMo.

**Weaknesses:**

1. The idea of introducing momentum methods into diffusion models for acceleration has been widely proposed and implemented [1-2], which affects the innovativeness of this paper to a certain extent.
2. The task scenarios do not cover other diffusion model tasks such as video generation and image editing, and the acceleration effect of FeMo in dynamic sequences or non-generative tasks remains to be verified.
3. Key parameters (e.g., $\beta$) need to be set empirically with defined ranges, and the adaptive mechanism is subject to manual constraints.

[1] Wang X, Dinh A D, Liu D, et al. Boosting Diffusion Models with an Adaptive Momentum Sampler[C]//IJCAI. 2024: 1416-1424.

[2] Wu Z, Zhou P, Kawaguchi K, et al. Momentum-accelerated Diffusion Process for Faster Training and Sampling[J].

**Questions:**

1. Tables 1 and 2 in the appendix show the experimental results of FeMo and Adapted-FeMo respectively, but the difference between the two is not significant under certain configurations (e.g., $\mathcal{N}=10, \mathcal{O}=1$ and $\mathcal{N}=3, \mathcal{O}=2$). How do the authors view the issue of Adapted-FeMo's ineffectiveness in such cases?
2. Some previous studies have shown that the sampling trajectories of diffusion models in 3D space are very similar.[1] Is this contradictory to the findings in Figure 2 of this paper?
3. Have the authors explored the maximum acceleration effect of FeMo and Adapted-FeMo?

[1] Defang Chen, Zhenyu Zhou, Can Wang, Chunhua Shen, and Siwei Lyu. On the trajectory regularity of ode-based diffusion sampling. arXiv preprint arXiv:2405.11326, 2024.

---

> ### Author Response · Authors · 2025-11-24
> **Feedback to Reviewer Vsqw**
>
> Thank you very much for your question; your suggestions have been extremely helpful to us. We have also included additional details, which are highlighted in blue in the revised paper.
>
> > **Q1: the issue of limited effectiveness.**
>
> **A1**: In the experiments on the base model DiT, under a relatively small acceleration ratio of $N = 3$, FEMO already achieves FID performance ($\text{FID} = 2.33$) that is very close to the unaccelerated DDIM-50 baseline ($\text{FID} = 2.32$), while providing a $2.77\times$ speedup. Under this setting, the adaptive mechanism further helps FeMo find the optimal weight for the historical momentum term.
>
> In the experiments on the base model FLUX with an extremely high acceleration ratio of $N = 10$, we observe a clear degradation in performance on DrawBench200. In this case, due to the large discrepancy between full-activation steps, the adaptive mechanism introduced for FEMO can no longer effectively capture these differences or help FEMO maintain prediction accuracy. Based on this observation, we plan to further explore more effective adaptive mechanisms for such ultra-high acceleration regimes in future work.
>
> > **Q2: sampling trajectory.**
>
> **A2**: Thank you very much for your question. The work you referred to shows that, for ODE-based diffusion models, the sampling trajectories exhibit a consistent regularity: regardless of the initial noise distribution or the target real data sample, all sampling paths follow an almost straight connection from the starting point to the final point. This conclusion is not in conflict with the visualization we present in Fig. 2.
>
> In Fig. 2, we show, for the DiT model, a comparison of sampling point trajectories for different samples at **DiT block index = 0** (left) and **DiT block index = 15** (right). This visualization is intended to illustrate our empirical finding that, across different samples, the output trajectories of the same DiT block index can evolve differently as the timestep changes. This, in turn, motivates our design: when generating different samples, the weight of the historical momentum should be adaptively adjusted so as to improve the prediction accuracy of FeMo under different generation tasks.
>
> > **Q3: The maximum speedup of FeMo.**
>
> **A3**: Thank you again for your question. The **maximum speedup of FeMo without causing degradation in generation quality** is achieved at **$N = 9$** on DiT with a **$7.10\times$** acceleration, and at **$N = 8$** on FLUX with a **$6.24\times$** acceleration. At the same time, we would like to emphasize that **Adapted-FeMo** is mainly designed to **improve quality preservation under a fixed acceleration ratio** by adaptively adjusting the weight of the historical momentum term. For example, on FLUX under the **same acceleration setting**, the previous baseline achieves its best **ImageReward$\uparrow$** score of **0.8396** (DuCa, $N = 10$), whereas **Adapted-FEMO** successfully improves this to **0.9501**.

---

> ### Comment · Reviewer_Vsqw · 2025-11-28
> **Reply to Author Rebuttal**
>
> Thank you for your response! Your reply has addressed some of my concerns. Considering the comments from other reviewers, I have decided to keep the score unchanged.

---

> > ### Author Response · Authors · 2025-12-03
> > **Feedback to Reviewer Vsqw(Second)**
> >
> > Thank you once again for the valuable time and effort you have dedicated to reviewing our work and for your careful evaluation of our experiments and metrics, as well as your suggestions on generality and limitations, which have guided us to conduct additional analyses and present our results more clearly.

---

### Official Review · Reviewer_yGZN · 2025-10-30

**Soundness:** 3
**Presentation:** 2
**Contribution:** 2
**Rating:** 2
**Confidence:** 3

**Summary:**

This paper proposes the Feature Momentum (FEMO) algorithm, which leverages a “cache-then-reuse” strategy based on the assumption that features from previous timesteps are similar to those in subsequent ones. The authors incorporate a momentum mechanism that predicts features across different timesteps, which they claim capture the temporal dynamics of diffusion models. Additionally, they propose Adapted-FEMO, an extension that adaptively searches for the optimal momentum coefficient for each generated sample. According to the authors, their approach achieves up to 7.10× speedup on DiT and 6.24× on FLUX.

**Strengths:**

The paper proposed a momentum mechanism to model the dynamics of diffusion models in different timesteps to predict features of diffusion models.

**Weaknesses:**

The paper needs to be rewrite to make sure clarity, especially the description of the algorithm. For example,
The sub-section: Feature Caching and predicting for Diffusion model did not provide reference.

**Questions:**

What does eqn. (3) means
Figure 4: what are (a) and (b) referred to in the caption, and the color used is too light. It is hard to read.
% in T%N is not an standard operator. What is the definition?
The authors stated that: "T is the full activation step closest to the first feature reuse step. It
typically does not directly equal the total number of timesteps"
Possible mistakes:
1. I believe there is a mistake in eqn. (1). It should be $\bar{\alpha}$, not $\alpha$, in order to have eqn. (2) follows. Equ. (1) is an approximate of the diffusion model. I suggest to use the SDE form for definition.
2. $\epsilon_t$ or  $\epsilon_{\theta}$

---

> ### Author Response · Authors · 2025-11-24
> **Feedback to Reviewer yGZN**
>
> Thank you very much for your thorough review of our manuscript. Our responses are provided below. We have also included additional details, which are highlighted in blue in the revised paper.
>
> > **Q1: What means of Eq.(3)?**
>
> **A1**: Eq. (3) gives the original baseline formulation for diffusion model acceleration based on feature caching. It indicates that the network output at timestep $t$ is stored in a cache and then reused over the following $N-1$ timesteps, thereby accelerating the inference process.
>
> > **Q2: About T.**
>
> **A2**: Here, `T % N` denotes the modulo operation, i.e., the remainder when $T$ is divided by $N$. We take DiT with the DDIM-50 sampler as an example to explain the statement that **“$T$ is the full-activation step closest to the first feature-reuse step; it usually does not directly equal the total number of timesteps.”** Concretely, sampling starts at $T = 49$ and the final output is obtained at $T = 0$. The Cache family of methods observes that, during the first two sampling steps, the similarity between adjacent timesteps is relatively low. Therefore, FeMo is only activated from timestep $T = 48$ onward to accelerate inference.
>
> About $\epsilon_t$ and $\epsilon_\theta$, $\epsilon_t$ represents a denoising network with inputs $x_t$ and $t$, and $\theta$ denotes the trainable parameters. The training process involves optimizing $\theta$ such that the predicted noise removal approximates the noise added during the forward process. Importantly, $\epsilon_\theta$ denotes a denoising network with its parameters $\theta$ that takes $x_t$ and $t$ as the input and then predicts the corresponding noise for denoising. This has been discussed in Section 3.1 Preliminary of the original paper.
>
> In addition, we have corrected other relevant details in the manuscript. Thank you again for your valuable question.

---

> ### Comment · Reviewer_yGZN · 2025-11-26
>
> Thanks to the authors for their response! Please explain this sentence in Page 4, line 202-203: "The input data $x_t$ consists of a sequence of tokens representing various patches within the generated images. This can be expressed as $x_t = \(x_i\)^{H×W}_ {i=1}$, where $H$ and $W$ correspond to the height and width of the images."
> My questions is: It seems to me that $X_t$ is the vector of pixel values of an image (can we just assume one image?).
> How the  vector $x_t$ (of pixel values) be consisted of a sequence of tokens representing various patches within the generated image (say one image)? If so, how many patches used for an image?
>
> Also, I noticed Eqn. (8) in Page 5 does not depend on $\beta$, while connection between the terms $\cal{M}$ and $\cal{F}$ functions are built via Eqn. (6), which depends on $\beta$. In addition, Eqn. (8) is not used anywhere. The authors should verify the content carefully and make it clear to the readers in the paper.

---

### Official Review · Reviewer_vmTb · 2025-10-31

**Soundness:** 3
**Presentation:** 3
**Contribution:** 2
**Rating:** 4
**Confidence:** 3

**Summary:**

This paper proposes a training-free acceleration framework for diffusion transformers, namely Feature Momentum (FEMO) and its adaptive variant Adapted-FEMO. FEMO leverages a momentum mechanism to predict future features from the derivatives of historical representations, thereby skipping redundant computations and significantly accelerating inference. Adapted-FEMO further introduces an adaptive weighting mechanism that dynamically adjusts the momentum coefficient based on the feature trajectory of each sample, improving robustness and generality. Experimental results show that FEMO and Adapted-FEMO achieve up to 7.10× acceleration and 6.24× on DiT and FLUX models while maintaining image quality, demonstrating their effectiveness and applicability in accelerating diffusion model inference.

**Strengths:**

1. The paper introduces the FEMO framework, which employs a momentum-based prediction mechanism to estimate future features and skip computation steps, achieving substantial inference acceleration.

2. Experiments show that FEMO and Adapted-FEMO can achieve high speedup ratios on FLUX and DiT models with almost no degradation in generation quality, indicating that they have high practical value and robustness.

3. The work provides an error-bound analysis and bias correction mechanism, combining theoretical reasoning with empirical validation to support the proposed approach.

**Weaknesses:**

1. **Scalability of the paper.** The experiments focus on visual diffusion models, such as DiT and FLUX, but there are relatively few experiments on other types or for different modalities. It is hoped that relevant experiments can be added to demonstrate the model's generality and scalability.

2. **Limited theoretical depth.** Although the paper provides error bound analysis, the discussion of the theoretical performance guarantees of FEMO and Adapted-FEMO in different models and tasks is not in-depth enough, such as the convergence and stability analysis of the momentum mechanism under different models.

3. **Lack of discussion on limitations** The paper does not adequately analyze potential failure cases. For instance, when feature trajectories change abruptly under high-noise or complex conditions, the momentum mechanism may mispredict feature directions, and the adaptive $\beta$ update may become unstable under distribution shifts. A more thorough discussion of these limitations would improve transparency.

**Questions:**

1. From a theoretical perspective, can the momentum mechanism in FEMO be interpreted as a first-order integral form of an ODE solver? If so, could this connection to numerical integration theory provide stronger mathematical support for the method’s design?

2. Does the $\beta$-update rule in Adapted-FEMO have convergence or stability guarantees? If $\beta$ is iteratively updated, could momentum drift or oscillation occur, and how might that affect robustness? Including a formal analysis or empirical validation would be valuable.

3. Regarding reproducibility: what are the additional computational and memory costs of FEMO? Does the caching process introduce significant overhead? Could the authors provide more details or visualizations on hyperparameter sensitivity and memory consumption?

---

> ### Author Response · Authors · 2025-11-24
> **Feedback to Reviewer vmTb**
>
> Thank you for your comprehensive and detailed review of our paper and the recognition of our work's clarity and effectiveness. We also add the experiment and details, and highlight using blue color in the paper. We provide our feedback as follows.
>
> > **Q1: Explanation of the integral formulation.**
>
> **A1**: Thank you for your question. In the following, we illustrate the argument using the application to DiT as an example. The cache-based acceleration family to which FEMO belongs is mainly designed to save computation time and FLOPs of the attention and MLP modules in each DiT layer across neighboring timesteps. Concretely, it caches the attention and MLP outputs of the DiT block at the previous timestep, then uses the FeMo algorithm to predict the attention and MLP outputs of the DiT block at the same index for the next timestep. At the last layer of the final DiT block, a `FinalLayer` consisting of layer normalization, conditional modulation, and a linear transformation is applied to obtain the current timestep’s $\epsilon_t$.
>
> Based on this working mechanism, we can see that the obstacle to explicitly representing the noise $\epsilon_t$ using FEMO lies in this final linear mapping layer. Moreover, the first-order integral property introduced by FEMO would be broken after passing through the `FinalLayer`.
>
> > **Q2: the stability of the adaptive mechanism.**
>
> **A2**: Thank you very much for your question. For the current expansion-term weight parameter $\beta$ in the adaptive mechanism, we start from theoretical analysis and then adjust it flexibly in practice. Regarding the selection strategy for Adapted-$\beta$, the scheme presented in the paper is our final choice. The reason is that we initially used a regression-based scheme: we let FeMo perform one additional feature prediction at the full-activation step (i.e., when $k = N$ in Eq. (8)), and then solved for the optimal $\beta$ in closed form based on the gap between the prediction and the “true” computed value. This, however, introduces non-negligible extra computational cost.
>
> To reduce this overhead, we examined the evolution of the adaptive weights produced by this regression-based scheme and observed that they are clearly constrained around $\beta_T$. Based on this empirical observation, we ultimately chose the simpler adaptive scheme described in the paper and accordingly set the value of $\gamma$ used in our experiments, which does **not** incur additional computational cost. As for parameter stability, we provide a detailed supplementary analysis in Section 5.2 of the paper.
>
> > **Q3: Regarding reproducibility.**
>
> **A3**:We report the peak GPU memory usage (GB) of FeMo and the baseline on DiT under identical configurations and experimental settings, as shown in the table below.
>
> |**Method (DiT-XL/2)**|**FLOPs(T)↓**|**Speed↑**|**FID↓**|**GPU memory usage(GB)↑**|
> |---|---|---|---|---|
> |*DDIM-50 steps*|23.74|×1.00|2.32|10.10|
> |*DDIM-25 steps*|11.87|×2.00|2.95|10.10|
> |*ToCa($\mathcal{N}$=13)*|4.03|×5.90|21.24|4.15|
> |*DuCa($\mathcal{N}$=12,$\mathcal{R}$=90%)*|3.94|×6.02|31.97|3.14|
> |*Taylorseer($\mathcal{N}$=7,$\mathcal{O}$=4)*|3.82|×6.22|3.59|4.66|
> |*Adapted-FeMo($\mathcal{N}$=7,$\mathcal{O}$=2)*|3.82|×6.22|**3.36**|11.25|
>
> Although our method introduces some additional memory overhead, it can still perform full inference on commonly used consumer GPUs in the market (e.g., RTX 3090 with 24 GB of memory), as illustrated in the figure. At the same time, we will further explore appropriate quantization techniques to eliminate redundant information in the cache, thereby minimizing memory usage while preserving generation quality.

---

> > ### Comment · Reviewer_vmTb · 2025-11-27
> > **Reply to Author Rebuttal**
> >
> > Thank you for the detailed rebuttal and the additional clarifications. After reviewing the responses, I acknowledge that some of my questions were partially addressed. However, my main concerns remain largely unaddressed:
> >
> > * **Q2 (adaptive β mechanism) and Q3 (memory usage)** were **partially addressed**. The authors clarified the design choices of the adaptive update and provided concrete memory statistics, which appropriately respond to these specific questions.
> >
> > * **Weakness on theoretical depth** was **only partially addressed**. The rebuttal explains why a clean ODE interpretation is difficult, but it does not provide additional theoretical guarantees (e.g., convergence or stability), so the core concern remains.
> >
> > * **Weakness on generality/scalability** was **not addressed**. No additional evidence or discussion was provided on applying FEMO beyond DiT/FLUX or to other modalities/architectures.
> >
> > * **Weakness on limitation/failure-case analysis** was **not addressed**. The rebuttal did not provide analysis or examples of scenarios where FEMO may fail.
> >
> > Given that the main concerns remain unresolved, I am keeping my original scores.

---

> > > ### Author Response · Authors · 2025-12-03
> > > **Feedback to Reviewer vmTb(Second)**
> > >
> > > Thank you once again for the valuable time and effort you have dedicated to reviewing our work and for your insightful comments on both the methodological design and the theoretical analysis, which have significantly helped us refine and strengthen the paper.

---

### Official Review · Reviewer_eFwD · 2025-10-31

**Soundness:** 2
**Presentation:** 3
**Contribution:** 2
**Rating:** 4
**Confidence:** 5

**Summary:**

The authors propose FEMO (Feature Momentum), a training-free method for cache-based sampling acceleration in Diffusion Transformers. This work improves upon the "cache-then-forecast" paradigm by introducing a momentum mechanism. Instead of relying on noise-sensitive, high-order derivatives like previous methods , FEMO predicts future features by calculating a stable, weighted trend of historical features, allowing computation steps to be skipped. The method achieves state-of-the-art acceleration compared to existing cache-based methods.

**Strengths:**

+ The method achieves a large speedups on Diffusion Transformers sampling acceleration compared to existing cache-based methods.
+ FEMO is a plug-and-play accelerator that can be applied to pre-trained models without any retraining or fine-tuning.

**Weaknesses:**

+ The paper's central claim to novelty is applying "Feature Momentum" for diffusion acceleration. However, the use of momentum in diffusion model acceleration is not new, and the paper fails to cite or discuss the extensive literature on this topic. While the authors may argue that FEMO applies momentum to *feature caching* rather than the *forward/reverse* process, these are deeply related concepts, as both leverage historical information throughout the diffusion process (forward or reverse). This omission prevents a clear differentiation of FEMO's contribution and significantly overstates its originality. For example:
  +  Sampling Acceleration:

  [1] Wizadwongsa, S., et al. "Diffusion Sampling with Momentum for Mitigating Divergence Artifacts." ICLR 2024.

  [2] Daras, G., et al. "Soft Diffusion: Score Matching with General Corruptions." TMLR.

  [3] Wang, X., et al. "Boosting Diffusion Models with an Adaptive Momentum Sampler." IJCAI 2024.

  + Diffusion Process Acceleration:

  [4] Dockhorn, T., et al. "Score-Based Generative Modeling with Critically-Damped Langevin Diffusion." ICLR 2022.

  [5] Wu, Z., et al. "Fast diffusion model." arXiv 2023.

+ The paper's title and claims are overly broad, suggesting a general method to "Accelerate Diffusion Transformers". However, the proposed method is specifically a **cache-based acceleration** technique, and its novelty is primarily within this sub-field. More, the experiments are made exclusively against other feature-caching baselines. The paper does not compare against other major acceleration families, such as fast samplers or model distillation. The claims and title should be refined to more accurately reflect this contribution.

+ The method's mechanism for momentum accumulation requires further justification. It builds a momentum term from a sparse history of computation steps (e.g., <6 steps in a 7.1x speedup setting), which differs significantly from traditional momentum (in optimizers or EMA) that relies on many steps for a stable estimate. The paper would benefit from a theoretical or empirical analysis of the stability and accuracy of this sparse-sampling approach.

+ The "Adapted-FEMO" variant introduces a notable number of hyperparameters (e.g., $\mathcal{N}$, $\mathcal{O}$, initial $\beta$, $\gamma$, and $\beta$-bounds) that require careful tuning, which could present challenges for reproducibility. There also appears to be a disconnect between the theoretical optimum derived for $\beta$ (Eq. 13) and the final implementation, which relies on a complex, heuristic-based adaptive search. It is unclear how this compares to prior momentum-based works [4, 5] that directly use the theoretical optimal coefficients.

+ There appear to be citation errors in the manuscript. For instance, the primary baseline "TaylorSeer" is repeatedly cited, but the corresponding bibliography entry seems to point to an unrelated paper ("Timestep embedding tells: It's time to cache for video diffusion model").

**Questions:**

see weakness

---

> ### Author Response · Authors · 2025-11-24
> **Feedback to Reviewer eFwD (1/2)**
>
> Thank you for your comprehensive review of our paper. We provide our feedback as follows. We also add the experiment and details, and highlight using blue color in paper.
>
> > **Q1: Comparison of novelty with prior related work.**
>
> **A1**: We thank the reviewer for the valuable comments and would like to further clarify FEMO’s relationship to prior work and its novelty. Existing methods [1–5] mostly introduce momentum into the **sampling process itself** (e.g., the updates of $x_t$, the noise, or the score / Langevin dynamics) to reuse historical information in the noise/state space. In contrast, FEMO does not modify the sampler; instead, it is built on a **feature-caching paradigm**, applying cross-timestep caching and momentum-style fusion to internal Transformer features of the diffusion model on top of a fixed sampler. Specifically, FEMO aggregates hidden features at the same DiT block across different timesteps to form a “feature momentum”, and adaptively adjusts the weight of historical features during inference to correct the current trajectory, thereby improving generation quality and stability **without changing the number of steps or the sampling rule**. This **online, feature-level adaptive momentum mechanism**, which controls how much historical information is used per sample and per timestep, is fundamentally different in operating space and form from prior approaches that only apply fixed or preset momentum coefficients in the state update equation. Our goal is therefore not to claim “the first use of momentum in diffusion models”, but to propose an adaptive, feature-caching–based acceleration framework that is orthogonal and complementary to existing fast sampling methods.
>
> > **Q2: About Model Distillation Methods.**
>
> **A2**: Thank you for your question. Feature-caching–based diffusion acceleration methods are typically used together with existing fast sampling methods, i.e., applied on top of them. Regarding distillation-based approaches, we also applied FeMo to a distilled model, and the experimental results are reported in the table. Specifically, the model is a 12B-parameter rectified flow Transformer distilled from FLUX.1 [pro] for 4-step generation. The results show that FeMo is compatible with distilled models and can achieve high acceleration rates with almost no performance degradation.
>
> |**Method (FLUX.1)**|**FLOPs(T)↓**|**Speed↑**|**Image Reward (DrawBench)↑**|
> |---|---|---|---|
> |**[dev]:50steps**|3719.50|×1.00|0.9898|
> |*50%steps*|1859.75|×2.00|0.9429|
> |*ToCa($\mathcal{N}$=3)*|1126.76|×3.30|0.9731|
> |*PAB($\mathcal{N}$=6)*|1744.62|×2.13|0.7518|
> |*DuCa($\mathcal{N}$=5,$\mathcal{R}$=90%)*|1078.34|×3.45|0.9896|
> |*DuCa($\mathcal{N}$=6,$\mathcal{R}$=90%)*|917.32|×4.05|0.9654|
> |**[mini]:50steps(ModelDistillation)**|979.50|×3.80|0.0168|
> |**[schnell]:4steps(StepDistillation)**|297.60|×12.50|0.9132|
> |*FeMo($\mathcal{N}$=2,$\mathcal{O}$=1)*|208.42|×17.90|0.9285|
>
> > **Q3: Further justification of the momentum mechanism.**
>
> **A3**:Thank you for your question. On the experimental side, we conduct extensive comparisons with FeMo’s baseline method, TaylorSeer, across all settings. The results demonstrate that FeMo can effectively leverage historical momentum terms to correct the prediction trajectory and achieve more accurate generation, from low acceleration $\mathcal{N}$ = 3 to high acceleration $\mathcal{N}$  = 9; please refer to Table 1 and Table 4 in the main paper for details. On the theoretical side, FeMo introduces an appropriate number of momentum terms under different acceleration ratios, thereby achieving lower prediction error compared with the baseline bound(Eq. (13) in the original paper):
>
> $E_m^{FeMo}(k) \leq \frac{(1-|\beta|)\sup_{\xi \in [t-k,t]} \|\Delta^m \mathcal{F}(x_{\xi})\|}{(m+1)!N^{m+1}} |k|^{m+1}+ \sum_{i=1}^m \frac{C_i}{i!}|k|^i|N|^{i-1}$

---

> ### Author Response · Authors · 2025-11-24
> **Feedback to Reviewer eFwD (2/2)**
>
> > **Q4: Comparison of novelty with prior related work.**
>
> **A4**:We thank the reviewer for the valuable comments on the hyperparameter design of Adapted-FEMO and the connection between theory and implementation. First, these hyperparameters ($\beta_0$, $\gamma$, $\lambda$, and the $\beta$-bounds, etc.) are all essentially defined along a **single dimension controlling the weight of historical features**, rather than forming a set of independent high-dimensional search variables. In the revised manuscript, we have added stability visualizations and sensitivity analyses, which show that the performance varies smoothly over a wide range of hyperparameter values and that the default configuration is sufficient to reliably reproduce our results. The visualization of our hyperparameter stability experiments is provided in Section 5.2. Second, the “theoretical optimal value” given by Eq. (13) is a reference coefficient derived under several idealized assumptions. Instead of fixing $\beta$ to this value, we use it as a theoretical prior to determine the search range and initialization of $\beta$, and then apply a simple adaptive mechanism to finely adjust it according to the nonlinear feature trajectories of different samples and timesteps. This differs from [4, 5], which directly adopt theoretically derived momentum coefficients in the sampling dynamics, and instead focuses on combining **theoretical priors with adaptive refinement** at the feature-caching level.
>
> In addition, we have corrected other relevant details in the manuscript. Thank you again for your valuable question.

---

> ### Comment · Reviewer_eFwD · 2025-11-27
>
> I thank the authors for their detailed responses. I have read the rebuttal and the comments from other reviewers (specifically Reviewer **Vsqw**, who shares similar concerns regarding novelty).
> Unfortunately, the rebuttal does not resolve my primary concerns regarding the novelty claims and the theoretical justification of the method.
>
> 1. The authors' rebuttal emphasizes that FEMO operates in feature space while prior works operate in sample space. I explicitly acknowledged and anticipated this distinction in my original review, where I wrote:
>
> > "While the authors may argue that FEMO applies momentum to feature caching rather than the forward/reverse process, these are deeply related concepts, as both leverage historical information throughout the diffusion process..."
>
> My stand is: mathematically, applying a momentum operator (accumulating historical gradients/differences to predict a future state) is a standard numerical technique. Merely shifting the application of this operator from the state variable $x_t$ (sample) to the hidden state $h_t$ (feature) does not constitute a fundamental methodological innovation sufficient to **ignore the extensive literature on momentum acceleration in diffusion**.
>
> 2. The authors' response regarding the "Adapted" mechanism confirms my concern about the stability of this approach: Momentum estimates require a sufficient history to stabilize (as seen in SGD or heavy-ball momentum). The authors admit that the theoretical optimal coefficient (Eq. 13) cannot be used directly and serves only as a "prior," requiring a heuristic search to work in practice. This validates my concern that the "momentum" estimate in this sparse-history regime (few steps) is too noisy to rely on the theoretical derivation, which can also evidence in performance in low acceleration.
>
> As the authors have confirmed that 1) the novelty relies on the distinction between feature/sample space (which I view as an implementation detail, not a fundamental shift) and 2) the theoretical derivation does not hold in practice without heuristic search, I maintain my original assessment.

---

> > ### Author Response · Authors · 2025-12-03
> > **Feedback to Reviewer eFwD(Second)**
> >
> > Thank you once again for the valuable time and effort you have dedicated to reviewing our work and for your detailed feedback on practicality, baselines, and efficiency, which has greatly helped us clarify our positioning and improve the experimental and implementation details.

---

### Meta-Review · Area_Chair_UXh9 · 2026-01-07

**Summary:**

This paper presents an acceleration framework for diffusion models by predicting future features from past steps.
The submission received mostly negative reviews from the reviewers.
The reviewers mainly recognize the performance gains, practicality and flexibility, and paradigm shift from simple caching.
The main concerns from the reviewers were limited novelty (eFwD, Vsqw), limited theoretical depth (eFwD, vmTb), limited generalization to other models and modalities (vmTb, Vsqw), unjustified momentum mechanism (eFwD), complexity of hyperparameters (eFwD, Vsqw), and reproducibility concerns (vmTb).
After reading the paper, the reviewers' comments and the authors' rebuttal, the AC believes the authors' responses would have partially addressed the reviewers' concerns, but there would still be outstanding concerns regarding limited novelty, shaky theoretical grounding, and shown limited generalization applicable to different models and modalities. The feedback from yGZN is downweighed due to insufficient constructiveness. The AC believes the remaining weaknesses would still outweigh the merits and does not recommend acceptance at this time.

**Reviewer Concerns:**

Reviewers' concerns mostly addressed:
- Reproducibility (vmTb)

Partially addressed concerns:
- Theoretical connections (eFwD, vmTb)
- Hyperparameter complexity (eFwD, Vsqw)
- Limited novelty (eFwD, Vsqw)

Outstanding concerns:
- Generalization to other models and modalities (vmTb, Vsqw)
- Justification on the mechanism of momentum (eFwD)

**Reviewer Scores:**

I think all reviewers would keep their original ratings.

---

### Decision · Program_Chairs · 2026-01-26

Reject